# The Wave Model of Sleep Dynamics and an Invariant Relationship between NonREM and REM Sleep

**Vasili Kharchenko** [1,2] **and Irina V. Zhdanova** [3,4,*]

1 Department of Physics, University of Connecticut, Storrs, CT 06269, USA; vasili.kharchenko@uconn.edu
2 Institute for Theoretical Atomic, Molecular & Optical Physics, Harvard-Smithsonian Center for Astrophysics, Cambridge, MA 02138, USA
3 Department of Anatomy and Neurobiology, Boston University School of Medicine, Boston, MA 02118, USA
4 BioChron LLC, Worcester, MA 01605, USA
* Correspondence: irina.zhdanova@bio-chron.com

**Abstract:** Explaining the complex structure and dynamics of sleep, which consist of alternating and physiologically distinct nonREM and REM sleep episodes, has posed a significant challenge. In this study, we demonstrate that a single wave model concept captures the distinctly different overnight dynamics of the four primary sleep measures—the duration and intensity of nonREM and REM sleep episodes—with high quantitative precision for both regular and extended sleep. The model also accurately predicts how these polysomnographic measures respond to sleep deprivation or abundance. Furthermore, the model passes the ultimate test, as its prediction leads to a novel experimental finding—an invariant relationship between the duration of nonREM episodes and the intensity of REM episodes, the product of which remains constant over consecutive sleep cycles. These results suggest a functional unity between nonREM and REM sleep, establishing a comprehensive and quantitative framework for understanding normal sleep and sleep disorders.

**Keywords:** sleep; cycles; architecture; dynamics; ultradian; NREM; REM; wave; mathematical model

## 1. Introduction

Despite significant progress in understanding the brain mechanisms involved in sleep regulation [1], the physiological function and dynamics of sleep within the sleep–wake homeostasis framework [2] remain uncertain and a topic of debate [3,4]. Sleep has a complex structure, known as sleep architecture, and, in humans, typically includes five to six sleep cycles per night (Figure 1a). Each sleep cycle consists of two types of sleep: it starts with nonrapid eye movement sleep (NREMS) followed by rapid eye movement sleep (REMS).

In contrast to the wake state, both NREMS and REMS are characterized by a low perception of the environment, but they are otherwise remarkably different states of the organism. Gradual disengagement from the environment in NREMS is associated with slow brain activity and reduced muscle tone, though capacity for motor activity is preserved. REMS is known as paradoxical sleep, since it manifests as high brain activity and rapid eye movements, active dream mentation, irregular heart rate and respiration, and sexual arousal, all on the backdrop of further reduction in perception and loss of muscle tone and thermoregulation [5,6]. This unique simultaneous presence of wake-like and sleep-like features makes the nature and significance of REMS particularly puzzling.

Understanding the intricate patterns of sleep dynamics represents a major challenge in the field. For several decades, there has been a recognized need for a comprehensive mathematical model that can describe both the sleep–wake and NREMS–REMS state transitions. Such a unifying model could provide a framework for elucidating the underlying mechanisms and help identify key factors and variables that affect the timing, duration, and quality of the sleep state and its components.

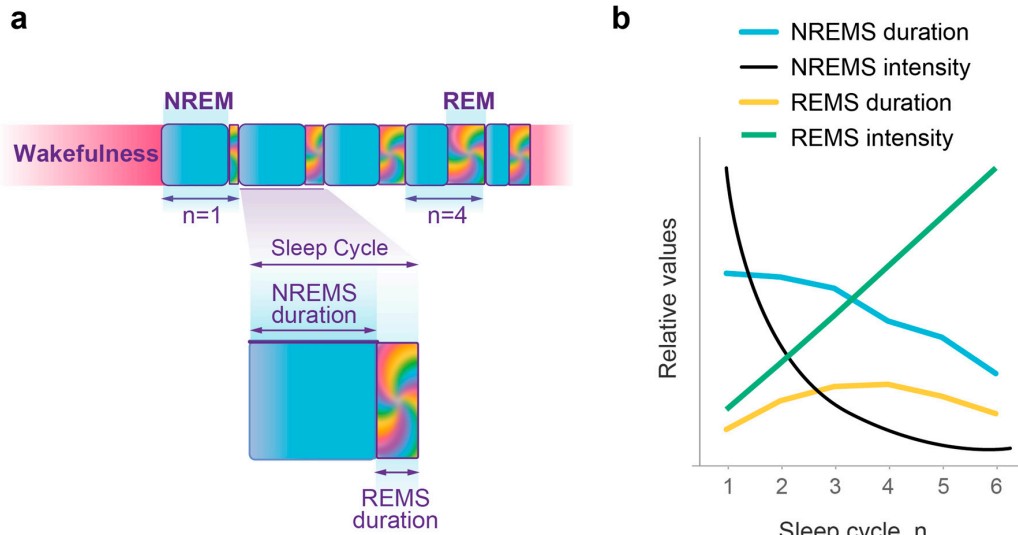

**Figure 1.** Human sleep architecture and overnight dynamics of the four primary sleep measures. (**a**). Schematic of consolidated sleep pattern, consisting of consecutive sleep cycles, each including longer NREMS (cyan) and shorter REMS (multicolor) episodes. Five sleep cycles are shown. (**b**). Schematic of typical changes in sleep measures over consecutive sleep cycles of regular nighttime sleep: NREMS duration (cyan), NREMS intensity (% of slow-wave sleep, black), REMS duration (orange) and REMS intensity (REM density, green).

A major advance was achieved by the two-process model of sleep regulation [2], which is based on the empirical analysis of the intensity of NREM sleep. It proposed that, during wakefulness, homeostatic mechanisms define an exponential rise in "sleep pressure" or "need for sleep". This process was also described as the rise in "wake state instability" [7], manifesting as an increase in sleepiness, performance deficits, and variability in motor and cognitive responses [8–10]. As sleep deprivation continues, subjective and objective symptoms of state instability diversify, involving autonomic regulation, microvascular and cardiovascular functions, and more [11–13]. The most prominent indicators of high wake state instability are microsleep intrusions and involuntary sleep initiations, which increase in frequency and duration over extended wakefulness [14]. Considering that the main driver of any state transition, whether in physics, chemistry, or biology, is the relative instability of one state compared to the other, these experimental observations suggest that the accumulation of wake state instability facilitates the spontaneous transition to a more stable sleep state.

These important models and concepts, however, did not incorporate REM sleep and, over the years, other theoretical approaches have been employed to address the complexities of the NREMS–REMS dynamic [2,15–18]. They considered the potential homeostatic relationship between REMS and NREMS or inter-REMS intervals [19], postulation of a specialized REMS oscillator that is active only during sleep [20], and modeling of ultradian oscillations based on classical mechanics of a particle oscillating between three potential wells representing wakefulness, NREMS, and REMS [16]. Although these models can simulate realistic patterns of sleep-like oscillations, they have not been able to quantitatively reproduce typical sleep architecture or functionally unite NREMS and REMS.

In our approach to sleep dynamics, we relied on earlier concepts of homeostatic regulation of sleep pressure and state instability [2,7]. We then suggested that quasi-periodic alternations between the two types of sleep may reflect the wave dynamics of the global physiological states of sleep and wake (Video Abstract). These global states are known to involve numerous interdependent negative feedback loops that define complex oscillations of critical biological variables around equilibrium setpoints on various levels, from the molecular to the systemic. To maintain the dynamic stability of an organism within physiological limits, it is essential for these homeostatic loops to be in coherence [21,22]. It

is well established in physics that coherent behavior of multiple oscillators typically leads to the formation of a wave; thus, global physiological states may present wave behavior. Wave processes are prevalent in biological systems, playing an important role in signal transduction and facilitating rapid long-range spatiotemporal coordination and signal preservation [23]. These attributes of wave processes can be critically important for global physiological states.

Since both sleep and wake states are stochastic processes, predicting the precise timing and duration of overall sleep or its components is challenging. However, on average, normal overnight sleep exhibits typical changes in the duration and intensity of NREM and REM sleep episodes (see Figure 1b). This pattern resembles the behavior of probability waves [24]. This would not be surprising, considering that both experimental and theoretical physics have shown that classical stochastic systems can mimic probability wave dynamics and be accurately described using wave mechanics [25–29].

Developing a comprehensive and unifying wave model of sleep dynamics requires incorporating the interactions between sleep and wake states as well as between NREM and REM sleep (Video Abstract). Such a model must then undergo rigorous quantitative testing of its predictions against the dynamics of multiple independent experimental measures representing both NREM and REM sleep. Notably, the dynamics of the four primary sleep measures are strikingly distinct from one another (see Figure 1b). Over successive sleep cycles, there is a gradual decline in NREM episode duration, accompanied by a rapid exponential decrease in NREM intensity, a linear increase in REM intensity, and a bell-shaped pattern of REM episode duration.

Here, we show that the wave model of sleep dynamics can accurately and quantitatively describe typical human sleep architecture and dynamics of all four primary sleep measures (Video Abstract). It then accurately predicts the effects of shorter- or longer-than-usual wakefulness on these sleep measures, resulting in relative "sleep abundance" or "sleep deficit", respectively. Importantly, the model suggests a functional link between NREMS and REMS, predicting an invariant relationship between NREMS duration and REMS intensity over the course of the night, which we now demonstrate experimentally.

## 2. Results

### 2.1. Modeling Sleep–Wake Homeostasis through Interacting Waves

We modeled the dynamics of wake and sleep states as an interaction of two waves, each shaped by the Morse potential well. This mathematical function is widely used in physics and chemistry to describe atomic interactions in molecules near their equilibrium, shaping the probability waves of molecular vibrations. Section 4 and Supplemental Figure S1 detail the choice of the Morse potential for modeling sleep dynamics, as well as some analogies between the dynamics of atomic and homeostatic states and their regulating parameters.

Figure 2 illustrates two distinct potential wells: $U_W$ representing the wake state and $U_S$ representing the sleep state. These potential wells effectively confine the respective waves and demonstrate their unique dependence on a single variable, the regulating parameter of state stability, $x$. In this context, $x$ serves as a one-dimensional reduction of a number of regulatory physiological parameters that define the system's capacity to maintain stability (see more on this in the Section 3). Theoretically, either state can manifest at any $x$ value, but their energy and thus stability exhibit distinct dependencies on $x$. Specifically, the wake state ($U_W$) exhibits minimal energy and maximal stability at lower $x$ value, compared to the sleep state ($U_S$). For each state, a deviation from its $x$ value of minimal energy increases the instability of the state. Therefore, as per the minimal energy principle, spontaneous selection between being in the wake or sleep state depends on the given $x$ value, with high $x$ values favoring the sleep state (Methods 1–3).

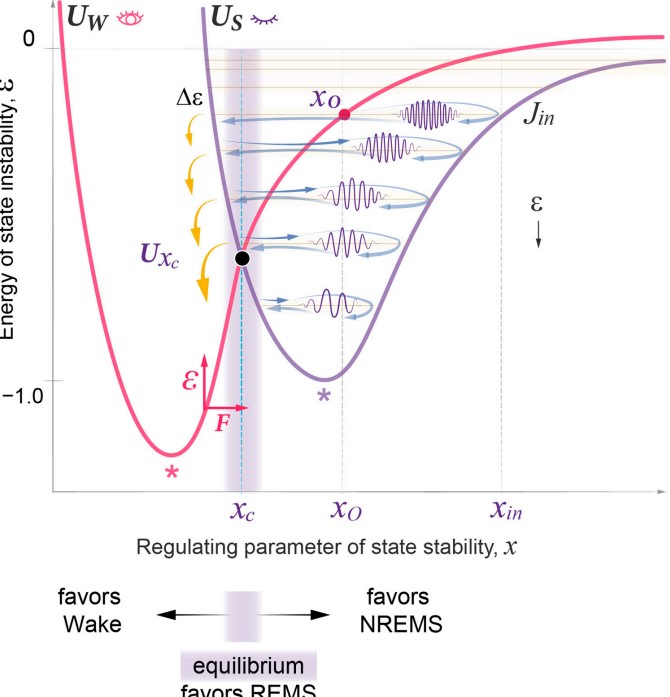

**Figure 2.** Wave model of sleep dynamics. See Video Abstract for dynamic presentation. In the wake state ($U_W$, red potential), driving force $F$ (horizontal red arrow) increases the value of regulating parameter $x$ (horizontal axis) beyond the sleep–wake homeostatic equilibrium setpoint $x_c$ (vertical cyan line) and the $x_c$ region of efficient interaction of the two states (gray area). This increases the energy of state instability $\varepsilon$ (vertical red arrow and vertical axis) above the homeostatic energy threshold ($Ux_c$, black dot; the crossing point of $U_W$ and $U_S$ curves). Sleep ($U_S$, purple potential) is initiated at $x_o$ (red dot) of the initial energy level $J_{in}$ (first sleep cycle). Relaxation of $x$ and $\varepsilon$ occurs in the form of a sleep wavepacket propagating along energy levels (blue arrows), where $x_{in}$—maximal deviation. When the wavepacket reaches $x_c$, a portion of its energy $\Delta\varepsilon$ (yellow arrows) is released. In the Morse potential, energy gaps $\Delta\varepsilon$ between levels increase linearly over the course of energy relaxation (downwards black arrow) and correlate negatively with the width of the potential well. *—maximal stability of the corresponding state. The direction of horizontal blue arrows shows which state is favored by changes in $x$ value: within the equilibrium $x_c$ region, REM sleep is favored, $x$-values greater than $x_c$ favor NREM sleep, and $x$-values less than $x_c$ favor the wake state. The schematic only serves to illustrate the concept. For actual position of $Ux_c$ and $J_{in}$ within $U_S$, see Supplemental Figure S2.

During daytime wakefulness, the collective action of homeostatic, circadian, and environmental forces drives $x$ away from the parametric region of a stable wake state, thus increasing the energy of state instability, $\varepsilon$ (Video Abstract). Once the system passes the $x_c$ equilibrium region, where the two curves cross and the energy of the two states is equal, the principle of minimal energy should allow for a spontaneous state transition from the wake to the sleep state. However, this is counteracted by the continuous work of the driving forces that promote consolidated daytime wakefulness and the further exponential rise in the energy of state instability.

At night, these driving forces subside, and the energy accumulated during wakefulness is spontaneously transferred from the unstable wake state to the more stable sleep state. Sleep onset occurs in between the two potential walls ($x_o$) at initial $U_S$ level $j_{in}$, generating a sleep wavepacket with energy $\varepsilon_{j_{in}}$ (Figure 2; Methods 4, 5). This event initiates the process of energy relaxation through a series of quasi-periodic oscillations, the sleep cycles, with each cycle resulting in a transition to a lower energy level. Eventually, this energy relaxation can move the system back to the original state of a stable wake state, closing the homeostatic sleep–wake loop.

Figure 3a illustrates the two simple model parameters—the width of the potential well ($\sigma$) and the number of the top occupied energy levels ($j_{in}$)—that are sufficient for the definition of the character of oscillations within the Morse potential, including the period, amplitude, and the energy lost during the transition to a lower level (energy gaps). The model predicts that these two parameters depend on two principal modulators of sleep dynamics—habitual sleep duration and the duration of wakefulness prior to the experimental sleep night. Thus, once the analysis of the dynamics of one primary sleep measure defines the values of these two model parameters for a given experimental group, they would allow us to predict the dynamics of other primary sleep parameters in the same dataset.

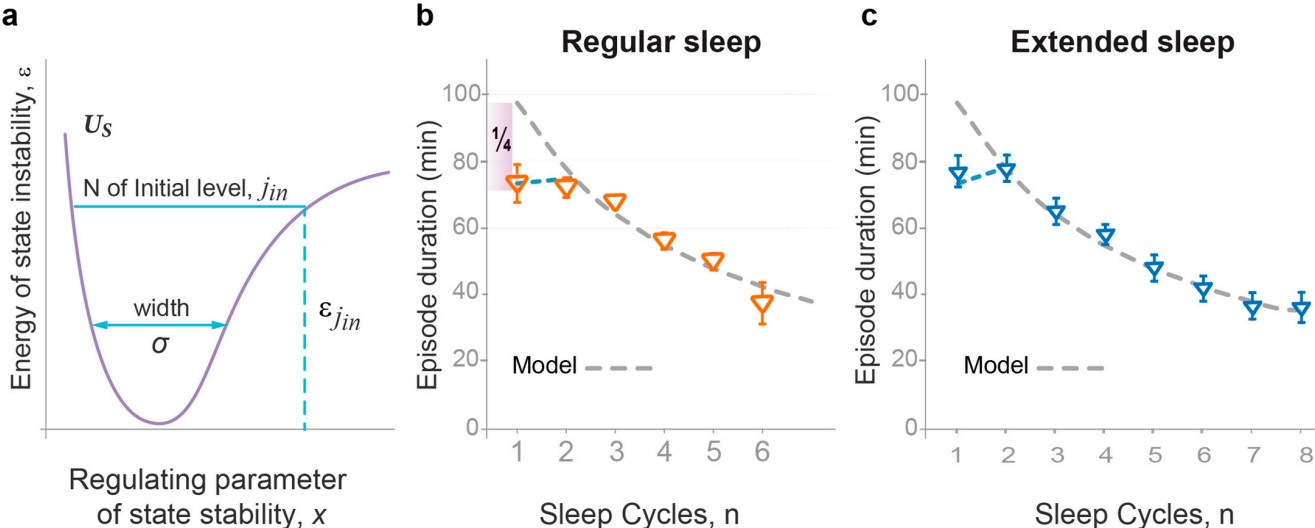

**Figure 3.** The wave model provides accurate quantitative description of dynamics in NREMS episode duration. (**a**). Two model parameters that allow us to describe NREMS episode duration (cyan solid line): $\sigma$—width of $U_S$ potential and $j_{in}$—order number (N) of the initial energy level at which sleep is initiated. $\varepsilon_{j_{in}}$ —energy of state instability reached by the time of sleep initiation (cyan dashed line). (**b**). Regular sleep: NREMS episode durations. Dashed line—changes predicted by the model, orange triangles—experimental data (min, mean ± SEM) as a function of sleep cycle order number n (horizontal axis). Due to $j_{in}$ being populated at a variable point $x_o$ (see Figure 3), the first experimental NREMS episode duration is, on average, $\frac{1}{4}$ shorter than the theoretical one for this energy level. Data collected in 24 young healthy subjects, 39 nine-hour nights (Method 16). $R^2$ value: 0.905. (**c**). Extended sleep. Dashed line—changes predicted by the model, blue triangles—experimental data (min, mean ± SEM) as a function of sleep cycle order number n (horizontal axis). Data collected in 11 young healthy subjects over 308 fourteen-hour nights, as reported by Barbato and Wehr [30]. First NREMS episode duration is explained in (**b**). $R^2$ value: 0.993.

## 2.2. The Wave Model Accurately Describes the Dynamics of NREMS Episode Duration

We used group data to test the model against physiological sleep measures as the stochastic nature of individual sleep patterns requires additional considerations, such as probabilistic treatment of model parameters and the influence of intrinsic and environmental perturbations or noise. First, we tested the model against the dynamics of NREMS episode duration in our group of young healthy volunteers with normal sleep of regular duration (Figure 3b, Method 16).

The decrease in NREMS episode duration over consecutive cycles (Figure 1b) is similar to the decline in the period of oscillations of the sleep wavepacket along the regulating parameter $x$ in the Morse potential (Supplemental Figure S2). As predicted, the right combination of the two model parameters $\sigma$ and $j_{in}$ (Figure 3a) allowed for the accurate depiction of relative changes in NREMS episode duration in this experimental dataset (Figure 3b; $\chi^2$ goodness of fit test $p > 0.88$, first episode excluded). As expected, the duration of the first NREMS episode was curtailed by approximately a quarter of the theoretically predicted whole period due to the position of sleep onset ($x_o$) in between the two potential walls (Figure 2).

To validate the model against independent observations and in subjects with different habitual sleep duration and prior wakefulness, we then tested the model against the dynamics of NREMS episode duration in a dataset collected by Barbato and Wehr [30,31]. In this experimental group, subjects displayed extended sleep duration over four weeks of daily exposure to 14 h sleep-favoring conditions. The model's predictions were again affirmed, providing its excellent fit to the experimental data (Figure 3c), with high statistical significance for this experimental group ($\chi^2$ goodness of fit test $p > 0.99$, first episode excluded) reflecting the high statistical power of this outstanding dataset (308 sleep nights). The model prediction that the first NREMS episode is, on average, curtailed by approximately a quarter of that predicted for the initial energy level was also confirmed in this analysis. Moreover, as predicted, longer sleep duration and more sleep cycles in the extended sleep group were associated with increased $\sigma$ and $j_{in}$ (Supplemental Table S1).

Collectively, our findings demonstrated that the overnight decline in the duration of NREMS episodes can be accurately and quantitatively described by the consecutive periods of wavepacket oscillations within the Morse potential based on the right combination of the two model parameters, $\sigma$ and $j_{in}$. However, for any model to claim adequate representation of an overall process, it is necessary to predict the dynamics of several independent measures characterizing this process that are not part of the development of the model. Hence, to demonstrate that the two interacting probability waves can indeed describe overall sleep architecture, the model had to precisely predict all four primary sleep measures (Video Abstract). To date, no sleep model has attained this validation threshold.

## 2.3. Quantitative Predictions of the Dynamics of NREMS Intensity

The intensity of a wave is directly proportional to the square of its amplitude $L^2$, as per wave mechanics. As we already knew the two model parameters $\sigma$ and $j_{in}$ for both the regular and extended sleep datasets (Figure 3), we calculated the amplitude of the wavepacket oscillations, L (Figure 4a; Method 13). The model also predicted that initial wave intensity and the rate of its decline positively correlate with the energy of the initial level, $\varepsilon_{j_{in}}$ . Accordingly, the intensity of wavepacket oscillation at each energy level was expressed as $\kappa L^2$, where the coefficient $\kappa$ is inversely proportional to $|\varepsilon|$, the absolute value of wavepacket energy (Method 13). Note that, in general, $\varepsilon$ values in a potential well are presented as negative, so an increase in $\varepsilon$ leads to a lower $|\varepsilon|$ (Figure 2).

Conventionally, NREMS intensity is evaluated based on the power of slow-wave activity (SWA) in the brain cortex or the duration of slow-wave sleep (SWS) [32]. To test our quantitative prediction regarding NREMS intensity, we compared SWS durations per cycle in experimental groups with regular and extended sleep to our model's predictions. Within each group, NREMS intensity was normalized to the first sleep cycle. We found that our theoretical curves were in good agreement with experimentally observed dynamics of NREMS

intensity for both experimental groups (Figure 4b, $\chi^2$ $p > 0.99$ for both datasets). Using SWA data for regular baseline sleep and recovery following 36 h sleep deprivation [32] produced similar results (Figure 4c, $\chi^2$ $p > 0.95$ for both datasets).

Overall, an accurate description of the dynamics of NREMS episode duration by a combination of two model parameters generated a precise prediction of the dynamics of NREMS intensity over consecutive sleep cycles. This also provided a mathematical and conceptual explanation for the expedient decline in NREMS intensity, which flattens out mid-sleep, substantially outpacing the decline in NREMS episode duration or overall sleep duration.

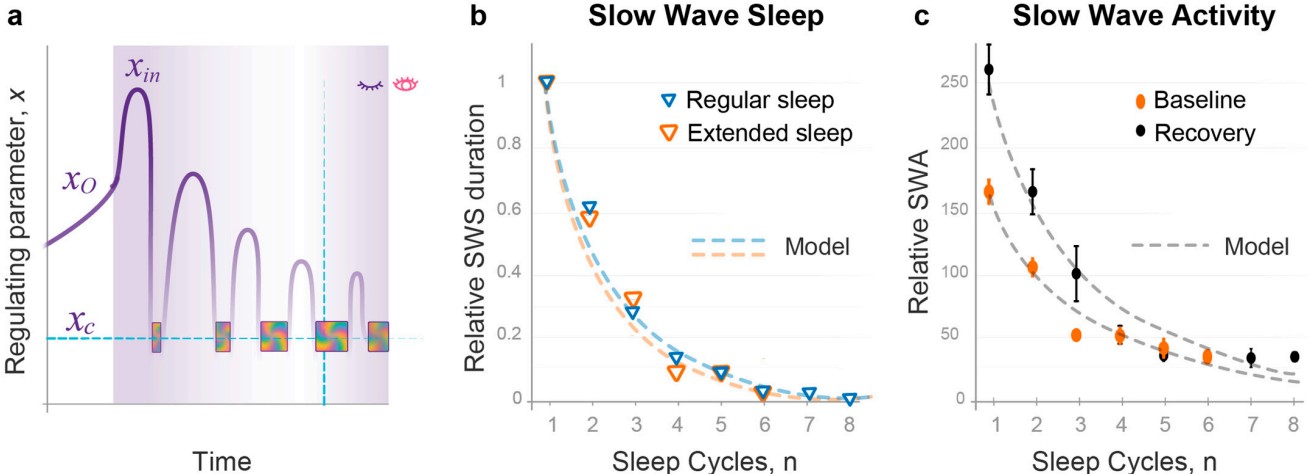

**Figure 4.** The wave model of sleep provides accurate quantitative prediction of dynamics of NREMS episode intensity. (**a**). Schematic of the time-dependent changes in the amplitude (L) of the regulating parameter of state stability, $x$ (vertical axis), during sleep (purple area). $x_o$—sleep onset, $x_{in}$—amplitude of the first cycle, $x_c$—equilibrium region of homeostatic threshold (horizontal dashed line), vertical dashed line—time of $Ux_c$ passage (see Figure 2), multicolor blocks—REMS episodes. (**b**). The decline in NREMS intensity is proportional to $L^2$. Theoretical curves (dashed lines) and experimental data (triangles) for the decline in NREMS intensity (proportional to $L^2$) over consecutive sleep cycles ($n$; horizontal axis) of regular sleep (orange, as in Figure 3b) and extended sleep (blue; as in Figure 3c; Barbato et al. [31]). Mean group data for slow-wave sleep duration are normalized to the first sleep cycle. $R^2$ values: regular sleep: 0.973; extended sleep: 0.990. (**c**). The decline in NREMS intensity during baseline sleep and following 36 h sleep deprivation, as assessed using slow-wave activity (SWA/cycle, %), is proportional to $L^2$. Theoretical curves (dashed lines) and experimental data (circles) for the decline in NREMS intensity over consecutive sleep cycles ($n$; horizontal axis) of regular baseline sleep (orange) and following 36 h sleep deprivation (black). Mean group data for slow-wave activity are from Dijk et al. [32]. $R^2$ values: baseline sleep: 0.862; recovery sleep: 0.814.

### 2.4. Quantitative Predictions of the Dynamics of REMS Intensity

Validation of the model's prediction that propagation of the wavepacket at each energy level corresponds to NREMS, as per both NREMS duration and intensity (Figures 3 and 4), suggested that the position of REMS episodes in between NREMS episodes may correspond to the transition of the wavepacket from a higher to a lower energy level. Accordingly, the intensity of REMS was then predicted to be proportional to the magnitude of the energy released during each transition (Figure 2). This corresponds to the energy gap $\Delta\varepsilon$ between adjacent levels that depends on the same two model parameters, $\sigma$ and $j_{in}$ (Figure 3a). Notably, in the Morse potential, as in other semi-classical potentials, $\Delta\varepsilon$ is inversely proportional to the period of oscillation and increases linearly towards lower energy levels.

As an experimental correlate of REMS intensity, we used the number of eye movements per minute of REMS episode, often referred to as REM density [33–36]. We elaborate on the choice of this measure in Section 3. Since energy gaps in the Morse potential are

small at high energy levels and increase linearly as the energy is relaxed (Figures 2 and 5a), the model predicted that REMS intensity should also be low at the start of sleep and increase linearly over consecutive REMS episodes. This is consistent with experimental data accumulated over the past decades that show linear increases in quantitatively assessed REM density over consecutive cycles of normal sleep [33–36] (Figure 5b). These increases depend on sleep homeostasis, not circadian regulation [36].

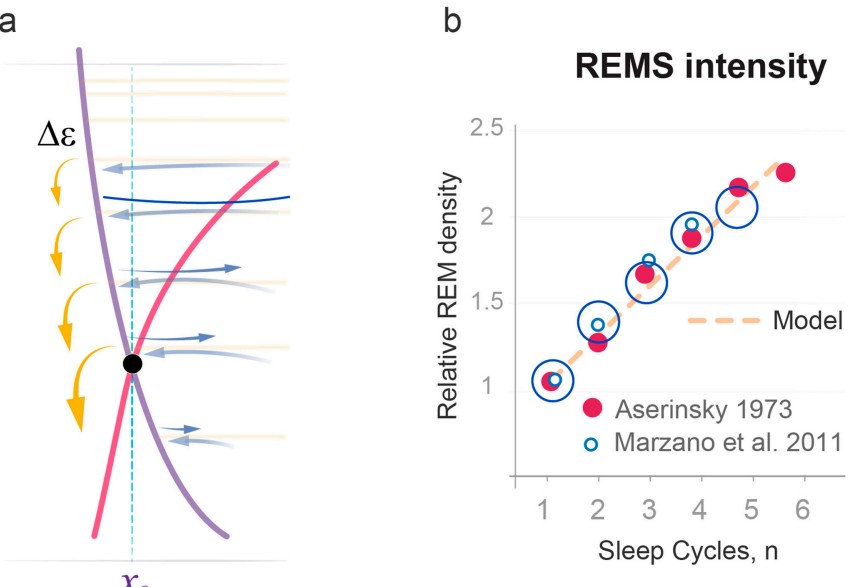

**Figure 5.** The wave model accurately predicts the slope of linear increase in REMS intensity. (**a**). Schematic of the region of homeostatic equilibrium between sleep and wake states (gray $x_c$ region in Figure 2, see caption for details) depicting linear increase in energy release (yellow arrows) that occurs in REMS due to linear increase in energy gaps between levels of Morse potential. (**b**). Rate of linear increase in REMS intensity (REM density) over consecutive sleep cycles documented in three similar but independent groups, as compared to theoretical prediction for a similar group by the wave model (orange dashed line). Group 1 (large open circle, n = 11 nights; $R^2$ 0.985; Method 16); group 2 in Aserinsky [34] (red circle, n = 20 nights, $R^2$ 0.961); group 3 in Marzano et al. [35] (small open circle, n = 50 nights; $R^2$ 0.997). Theoretical prediction was based on increase in energy gaps $\Delta\varepsilon$ over consecutive cycles, with no adjusting parameters used (n = 39 nights, as in Figure 3b). Mean group data for REM density are normalized to the first sleep cycle.

A strong validation of our model was then provided by comparing the theoretical and experimental slopes of linear increase in REMS intensity over regular sleep period (Figure 5b). In the model, the slope is defined by the same two model parameters $\sigma$ and $j_{in}$ (Figure 3a). The model also predicts that $j_{in}$ positively correlates with the duration of prior wakefulness, while $\sigma$ positively correlates with habitual sleep duration (Method 14). Together, this suggested that groups with similar sleep durations prior to and over the experimental period should have similar REMS intensity slopes.

To test this prediction, we compared the dynamics of quantitatively assessed REM density data for normal sleep of regular duration documented in three groups of young healthy volunteers, studied by us (n = 11 nights; Method 16), Aserinsky [34] (n = 20 nights), and Marzano et al. [35] (n = 50 nights). Figure 5b illustrates that the slope angles were almost identical between these independent experimental groups and matched the slope predicted by the model well (Method 16; $\chi^2$ $p > 0.99$ for all datasets). The latter theoretical slope was based exclusively on the values of $\sigma$ and $j_{in}$ documented in the analysis of NREMS episode durations during 39 nights of regular sleep (Figure 3b).

*2.5. Sleep Cycle Invariant: Theoretical Prediction and Experimental Demonstration*

Figure 6 and Video Abstract illustrates the wave model's prediction of a sleep cycle invariant (SCI) based on the validated mathematical description of NREMS and REMS dynamics. As described above, within each sleep cycle, NREMS episode duration corresponds to the period of oscillation of the sleep wavepacket, and this period is inversely proportional to the energy gap between levels, $1/\Delta\varepsilon$ (Method 5). On the other hand, the intensity of the subsequent REMS episode is directly proportional to the energy released at the end of the cycle, $\Delta\varepsilon$ (Figure 5). This predicts that the product of NREMS episode duration and the intensity of the following REMS episode should remain constant over consecutive sleep cycles, which constitutes the SCI (Figure 6; Method 5, 14, 15).

We suggested that various factors that interfere with sleep quality can modify SCI stability and therefore tested this prediction against high sleep efficiency data (not less than 93%, n = 11 nights). The product of NREMS duration and REMS intensity was calculated for each sleep cycle of each individual night to evaluate the SCI.

Despite NREMS duration and REMS intensity showing distinct dynamics over consecutive sleep cycles, the product of these two sleep measures (SCI) remained near constant over the course of the night (Figure 7a,b, *p* value > 0.92, see Statistical analysis in Methods). This confirmed the invariant relationship between NREMS and REMS over consecutive sleep cycles of normal high-efficiency sleep.

We then aimed to evaluate the SCI using previously published data on normal sleep and sleep disorders, but encountered a methodological challenge. A robust linear increase in REM density over the sleep period was originally demonstrated by Aserinsky, the co-discoverer of REM sleep [5], through quantitative assessment of rapid eye movements and was confirmed by others, also establishing its independence from the circadian phase [34–36]. Conversely, most studies reported REM density based on more technically simple semi-quantitative evaluations, such as assigning a score to a range of REMs or counting a number of intervals that contained REMs. These semi-quantitative measures can reveal major pathological changes in REMS intensity [37,38] but lack sensitivity, especially at high intensity levels [39], and thus can obscure normal linear patterns. Consequently, we could identify few studies that reported quantitative data on REMS intensity, and none that reported them over consecutive sleep cycles in parallel with NREMS episode durations.

The model predicted that the remarkable similarity in the dynamics of NREMS duration (Figure 3a,b) and quantitatively assessed REMS density (Figure 5b) among independent groups of young healthy subjects is indicative of similar dimensions of their sleep potential wells ($U_S$), specifically the wells' widths, and energy levels at which sleep is initiated. This further suggested that the SCI might be discernible even if NREM and REM data originate from distinct studies. We then used NREMS data from one study [30] and REMS intensity data from another [34], with both studies being of the highest quality and allowing their subjects the opportunity of extended sleep. Figure 7c illustrates the results of such a "hybrid" analysis that supports the model's prediction, demonstrating the SCI and further underscoring the inherent conservation of typical human sleep architecture, at least in healthy young adults.

In contrast, when we used the results of a semi-quantitative assessment of REMS intensity in combination with NREMS episode duration documented within one top-quality large-scale study [31], the invariant relationship was obscured (Figure 7d). Together, these results further supported the existence of the SCI in normal sleep and underscored the critical importance of the quantitative assessment of REM density.

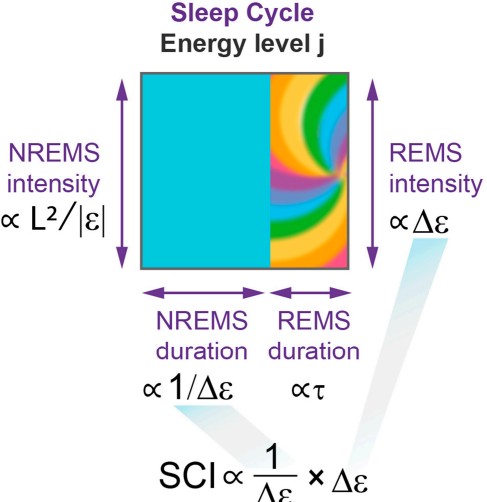

**Figure 6.** Wave model of sleep predicts sleep cycle invariant. Experimental sleep measures (purple font) are proportional ($\propto$) to the wave model parameters (black font). NREMS duration is $\propto$ to the period of $x$ oscillation and thus inverse $\propto$ to the energy gap, $1/\Delta\varepsilon$. NREMS intensity is $\propto$ to $L^2/|\varepsilon|$, where L is the amplitude of $x$ oscillation and $|\varepsilon|$ is the absolute value of wavepacket energy. REMS duration is $\propto$ to $\tau$, the lifetime of coherent superposition. REMS intensity is $\propto$ to energy gap, $\Delta\varepsilon$. The model predicts that the sleep cycle invariant (SCI), being the product of NREMS duration and REMS intensity, should remain constant over consecutive sleep cycles. Cyan—NREMS, multicolor—REMS.

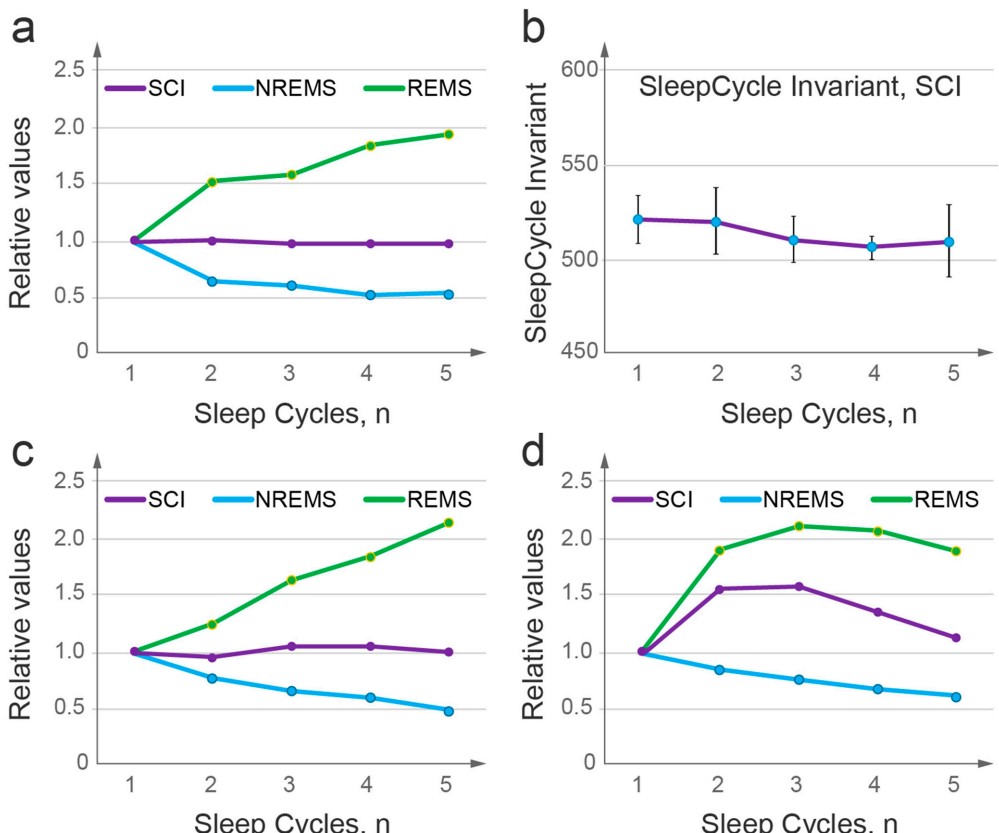

**Figure 7.** Sleep cycle invariant unites NREM and REM sleep. (**a**). The product of NREMS episode duration and REMS episode intensity—the sleep cycle invariant (SCI, purple). NREMS duration (cyan) and REMS intensity based on quantitative evaluation of REM density (light green) over consecutive cycles of regular sleep in a group of 11 young healthy volunteers with sleep efficiency of at least 93%; n = 11 for cycles 1–4 and n = 5 for cycle 5 (see Section 4). Data normalized to the first

sleep cycle (=1) for each parameter and presented as group means ± SEM per cycle *. (**b**). Variation in SCI over consolidated high-quality sleep. Data are presented as mean ± SEM; dashed line defines the mean value (515.2); the rest as in (**a**). (**c**). Analogous to panel (**a**), the plot shows SCI based on NREMS duration per sleep cycle, as reported by Barbato and Wehr [30] (n = 308 nights; 11 young healthy volunteers), and REMS intensity per cycle, as reported by Aserinsky [35] and evaluated using quantitative method (n = 20 nights; 10 young healthy volunteers). (**d**). Loss of linear time- and cycle-dependency of REMS intensity when assessed using semi-quantitative method (dark green) results in obscured NREMS–REMS invariant relationship (purple). REMS intensity was assessed in parallel with NREMS duration, as reported by Barbato et al. [31] (n = 208 nights; 8 young healthy volunteers). Data normalized to the first sleep cycle (=1) for each parameter and presented as group mean per cycle *. * In (**a–d**), the duration of the first NREMS episode was multiplied by 3/4, as per the model prediction of incomplete period of first cycle (see Methods and Figures 2 and 3).

### 2.6. Quantitative Predictions of the Dynamics of REMS Duration

Finally, we addressed the dynamics of the fourth primary sleep measure—the duration of REMS episodes. The validated model prediction that REMS intensity is directly proportional to the energy released during the transition of a wavepacket to a lower level suggested that the duration of REMS episodes is determined by the duration of this energy release process.

As the wavepacket approaches the wake state ($U_W$) towards the end of each NREMS episode, it enters the region of sleep–wake homeostatic equilibrium ($x_c$ region) where the interaction between sleep and wake waves is increased (Figures 2, 5a and 8). Such interaction typically delays the waves' propagation (Methods 6–9), leading to a temporary coherent superposition of two waves representing different states. The sleep–wake interaction is strongest at the crossing of the two potentials. This can create a resonant state that further enhances the strength of interaction between sleep and wake waves and prolongs the delay in wavepacket propagation within this $x_c$ region. The effect is maximal at resonance level $j_c$, the energy of which is distinct from $U_S$ levels and corresponds to the homeostatic energy threshold, $U(x_c)$ (Figure 8; Video Abstract, Methods 8–10). The energy dependence of the time that the wavepacket spends in the $x_c$ region of homeostatic equilibrium is then defined by the classical Lorentz resonance curve (Methods 11).

The same two model parameters, $\sigma$ and $j_{in}$ (Figure 3a), characteristic of a specific dataset, predicted energy relaxation dynamics as a function of cycle number, $n$. Fitting the right combination of resonance energy level and resonance width (Figure 9a) then allowed for a quantitative description of the nonlinear dynamics of REMS episode duration (Method 12).

This analysis was applied to both regular and extended sleep datasets (same as in Figure 3b,c and Figure 4b), and the model accurately described both patterns of REMS episode duration (Figure 9b,c; $\chi^2$ goodness of fit test $p > 0.95$ and $p > 0.99$, respectively). Under normal conditions, with no sleep debt accumulated prior to the experimental procedures or during the study, the dynamics of REMS episode duration follows a bell-shaped curve (Figure 9b,c). These experimentally observed bell-shaped patterns of REMS duration were consistent with the Lorentz resonance curve, according to which the closer the energy of the wavepacket is to the energy of resonance level $j_c$, the longer the presence of the wavepacket in the homeostatic equilibrium region, and thus the longer the duration of the REMS episode. Accordingly, maximal REMS duration was observed around the homeostatic energy threshold, $Ux_c$. The lower peak of the bell-shaped curve in extended sleep (Figure 9c) was consistent with wider resonance in a wider potential well that had more energy levels and smaller energy gaps (Supplemental Table S1).

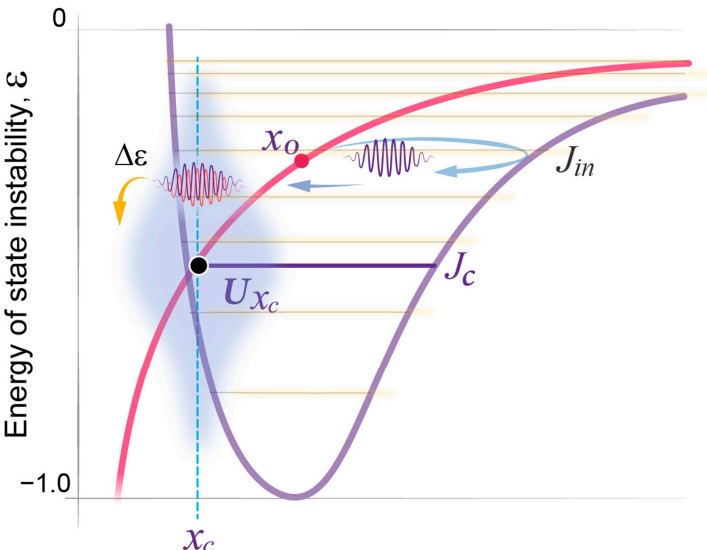

**Figure 8.** The duration of coherent superposition within the equilibrium region of sleep and wake states is enhanced by resonance. Around the crossing point $U(x_c)$ of $U_S$ (purple potential) and $U_W$ (red line), a strong sleep–wake interaction creates resonance level $J_c$ and resonance state (gray area) with energy-dependent ($\varepsilon$) strength (width of gray area). The propagation of the sleep wavepacket (purple) is temporarily delayed by the resonance within the $x_c$ equilibrium region. There, the wavepacket incorporates the sleep and wake waves and forms their coherent superposition (red-purple wavepacket). This represents REM sleep, associated with the release of a portion of energy $\Delta\varepsilon$ (yellow arrow). This is a schematic representation; for actual typical position of $J_c$, see Supplemental Figure S2.

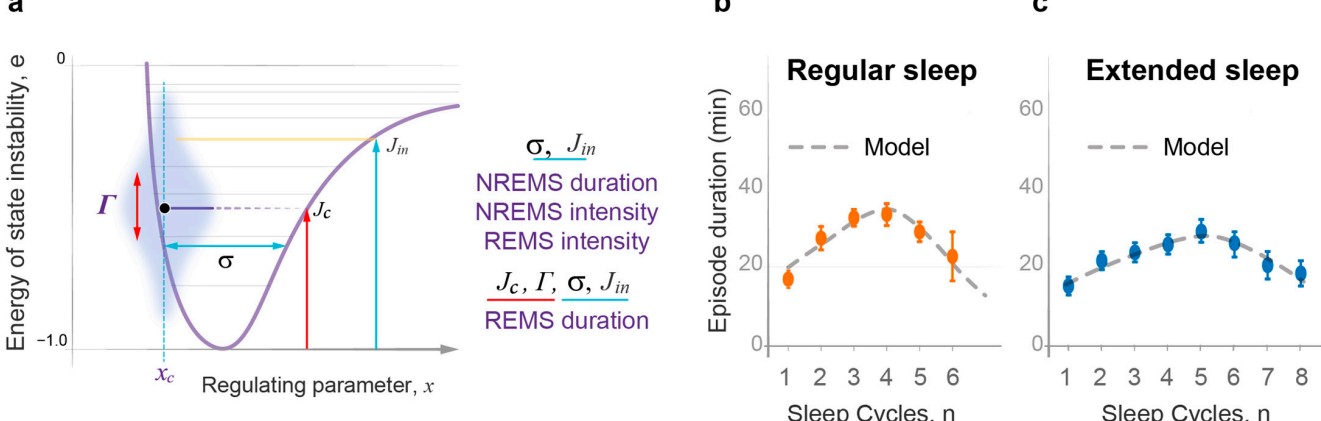

**Figure 9.** Accurate quantitative prediction of the durations of REMS episodes by the wave model of sleep dynamics. (**a**). In addition to model parameters $\sigma$—width of $U_S$ potential (cyan horizontal arrow) and $J_{in}$—initial energy level at which sleep is initiated (yellow line; and its energy—vertical cyan arrow) used in the analysis of the duration of NREMS episodes, intensity of NREMS and REMS (see Figures 4–6), quantitative description of REMS episode duration requires two more model parameters: $\Gamma$—resonance width (red vertical double arrow) and energy of resonance level $\varepsilon_{j_c}$ (red vertical arrow). Gray area—resonance region, with wider area depicting stronger resonance. Black dot—homeostatic energy threshold $U(x_c)$. (**b**). Regular sleep: theoretical curve (dashed line) and experimental data (min, mean ± SEM) for REMS episode durations (circles) as a function of sleep cycle order number n (horizontal axis). Data collected in 24 young healthy subjects, 39 nine-hour nights (Method 16). $R^2$ value: 0.811. (**c**). Extended sleep: theoretical curve (dashed line) and experimental data (min, mean ± SEM) for REMS episode durations (circles) as a function of sleep cycle order number n (horizontal axis). Data collected in 11 young healthy subjects, 308 fourteen-hour nights, as reported by Barbato and Wehr [30]. $R^2$ value: 0.965.

### 2.7. Model Predicts the Effects of Sleep Deprivation or Abundance on Sleep Architecture

Validating a model under perturbed conditions is essential for ensuring its robustness and reliability. In this report, we outline the model's predictions only for acute perturbations under entrained conditions when sleep is initiated at the habitual bedtime. These conditions provide a stable adaptive synergy between circadian and homeostatic regulation of sleep. The wave model demonstrated high accuracy in predicting the effects of acute sleep deprivation (causing sleep deficit) and shorter-than-usual wakefulness (causing relative sleep abundance) on sleep architecture, which have been extensively documented in various experimental settings [10,32–34,40].

Specifically, the model predicted that recovery sleep following acute moderate sleep deprivation starts with higher NREMS intensity due to higher initial energy and higher amplitude of wavepacket oscillations at higher levels, on which NREMS intensity depends (Figure 4). Figure 10 illustrates that the higher wake state instability reached following prolonged wakefulness is predicted to correspond to weaker resonance, and hence shorter initial REMS episode duration. Smaller energy gaps $\Delta\varepsilon$ between higher energy levels of the Morse potential result in lower initial REMS intensity (Figure 5). Initial NREMS episode duration is a less predictable parameter following sleep deprivation, mainly due to increased variation in $x_o$, the position of spontaneous sleep onset (Figure 2), or the occurrence of exceptionally long NREMS episodes due to the well-known "skipped REMS episode" phenomenon. The latter can be explained by very weak resonance at high energy levels, which leads to a very brief coherent superposition and thus to REMS duration that is too short to be reliably documented.

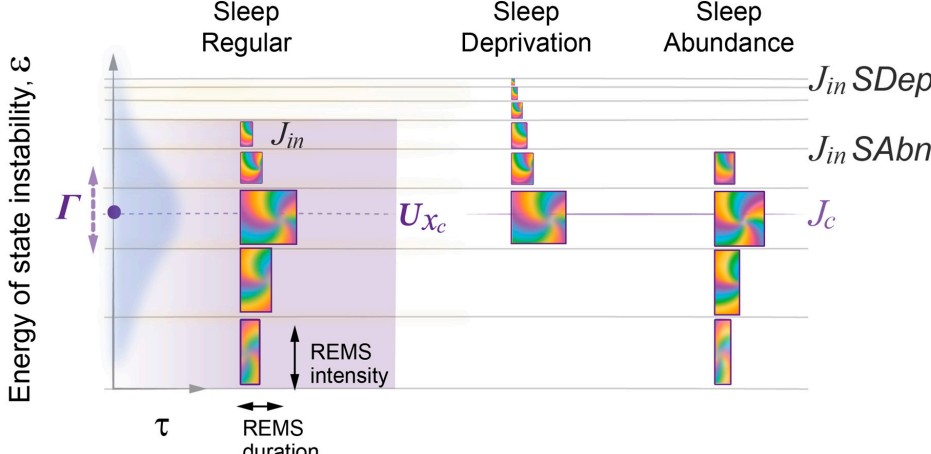

**Figure 10.** The wave model predicts the effects of sleep deprivation and sleep abundance on the duration and intensity of REMS episodes. *Regular sleep:* Maximal REMS duration (width of the multicolor blocks) occurs near resonance level $J_c$ at the homeostatic energy threshold ($Ux_c$, black dot, dashed line). Reductions in this experimental measure above and below the resonance level (gray area) are defined by resonance width ($\Gamma$, vertical double-arrow). REMS episode intensity (height of the multicolor blocks) follows a linear increase in energy gaps between levels (horizontal gray lines) of the Morse potential. *Sleep deprivation (SDep):* following a prolonged wake state, sleep starts at a higher initial energy level ($J_{in}$ SD), resulting in smaller initial duration and intensity of REMS episodes. If recovery sleep is of regular duration, awakening may occur before reaching or at the homeostatic threshold level, preventing the formation of a bell-shaped pattern of REMS episode duration. *Sleep abundance (SAbn):* following a brief wake state, with no prior sleep deficit, sleep is initiated at a lower initial energy level ($J_{in}$ SA), increasing the initial duration and intensity of REMS episodes. Recovery sleep can be shorter than usual due to the relaxing energy of instability reaching the homeostatic energy threshold faster than in regular sleep.

Another important prediction of the model for the effects of sleep deprivation is that recovery sleep of regular duration initiated at the habitual bedtime may not have enough time to reach below the homeostatic energy threshold and thus the resonance level. As a result, the REMS duration curve would not manifest itself in a bell shape but would only show a continuous increase (Figure 10, SDep).

In contrast to the effects of sleep deprivation, when sleep onset at the habitual bedtime follows prior acute sleep abundance, such as due to a prolonged daytime nap, sleep initiation would be associated with lower energy level, larger $\Delta\varepsilon$, and a smaller difference between $j_{in}$ and $j_c$ (Figure 10, SAbn). These accurately predict that sleep would start with a lower NREMS intensity, higher REMS intensity, and longer REMS episode duration, respectively.

## 3. Discussion

### 3.1. Interaction of Two Probability Waves Defines Dynamic Sleep Architecture

This study introduces an innovative conceptual model for understanding sleep dynamics, based on the interaction of two probability waves representing sleep and wake states (Video Abstract). The model is rigorously tested against experimental data that capture normal sleep patterns of both regular and extended duration. The results demonstrate a high level of statistical significance in the model's quantitative predictions of the dynamics of both the durations and intensities of NREM and REM sleep. This finding suggests an intrinsic connection between these two types of sleep, a concept that has been pursued for seven decades since the discovery of REM sleep [5]. This suggestion gains further support through experimental validation of the sleep cycle invariant predicted by the model. Notably, the discovery of invariants in various scientific fields has led to the development of unifying theories that explain diverse phenomena under a common framework.

The model relies on advanced mathematical tools from wave mechanics, yet it provides straightforward analytical formulas for key sleep measurements, enabling efficient analysis of extensive datasets. This simplicity arises from the minimal number of model parameters and the direct relationship between four specific characteristics of the quasi-periodic process within the Morse potential and the four primary sleep measures of each sleep cycle. These measures encompass the period of oscillation (NREM duration), its amplitude (NREM intensity), the energy gap between levels (REM intensity), and the time involved in overcoming that gap (REM duration).

The straightforward relationship between these four elements of sleep architecture implies that the sequential occurrence of NREM sleep followed by REM sleep within each sleep cycle reflects their distinct yet complementary roles. The model proposes that during NREM sleep, the evolution of the wavepacket prepares energy for release. As the system enters the equilibrium region between sleep and wake states, the strong interaction between the two waves results in their coherent superposition, leading to REM sleep. During REM sleep, a portion of the primed energy is released, followed by further evolution of the wavepacket during the subsequent NREMS episode. In this way, throughout successive sleep cycles, the gradual dissipation of the energy associated with state instability brings the system closer to a more stable wake state. Consequently, longer and more intense periods of REM sleep increase the likelihood of spontaneous awakening [31], as they facilitate a more effective reduction in state instability.

The description of REM sleep as a coherent superposition of two waves representing sleep and wake states may initially appear unexpected. However, this concept aligns precisely with the accurate quantitative predictions of the model. Importantly, there is a striking similarity between these phenomena. In physics, the coherent superposition of interacting waves can give rise to a distinct new state that simultaneously exhibits the characteristics of both constituent states, rather than being a mere combination or alternation between them [24,41]. This observation aligns with REM sleep, which simultaneously displays features associated with both sleep and wake states. Due to these unusual charac-

teristics, REM sleep is also conventionally referred to as "paradoxical sleep" [42] or a "third state" of consciousness [43], highlighting its unique nature.

### 3.2. REM Sleep Intensity and the Role of Sleep Cycle Invariant

The traditional diagnostic criteria for assessing normal REM sleep involve the presence of REMs, desynchronized EEG patterns, and the absence of skeletal muscle tone. The search for EEG correlates of REMS intensity has yielded inconclusive results. In contrast, changes in REM density or intensity (i.e., the magnitude of a quantity per unit) have emerged as a distinct feature of sleep dynamics [33–36]. Interestingly, this measure was not universally recognized as REMS intensity because, in analogy to the high NREMS intensity observed after prolonged wakefulness, it was expected that REMS intensity would also be high at the beginning of sleep, which sharply contrasted with the dynamics of REM density.

Eugene Aserinsky, the co-discoverer of REM sleep [5], conducted an extensive characterization of REM density across sleep periods of varying lengths using a precise quantitative method that involved counting each rapid eye movement [34–36]. He determined that REM density increases throughout the duration of sleep and proposed that it serves as a measure of "sleep satiety" [33]. This concept finds support in the wave model, which links REMS density, as an indicator of REMS intensity, to the amount of energy of instability relaxed within each sleep cycle. This energy release, as described by the Morse potential, exhibits a linear increase during the energy relaxation period. Importantly, the slope of this increase quantitatively aligns with that of REM density across consecutive sleep cycles (Figure 5b).

The model's prediction of the sleep cycle invariant and our empirical evidence supporting it (Figure 7a,b) lend new significance to REMS intensity, providing compelling evidence that NREMS and REMS are integral parts of a unified process. Discovering the SCI opens up new avenues for experimentation and conceptual analysis. In physics and other fields, invariant or symmetry relationships are generally indicative of conservation laws and principles governing a behavior, and commonly provide insight into the underlying mechanisms of a system. Similarly, the SCI arose from a general property of quasi-classical potentials, where the period of wavepacket oscillations is inversely proportional to the gap between energy levels, with the period and energy lost correlating with NREMS duration and REMS intensity, respectively. The existence of an invariant relationship between NREMS and REMS strongly suggests that these two types of sleep are not independent but rather regulated by a common mechanism.

The SCI may provide a unique metric that integrates NREMS and REMS to measure sleep quality for each cycle and across the entire sleep period. This presents a significant opportunity to investigate the dynamic mechanisms underlying sleep disturbances, such as the different types of insomnia, and provide new insights into the pharmacodynamics of both established and novel medications, as they exhibit unique, time-dependent effects on sleep.

The SCI may be particularly relevant in the context of psychiatric and neurological disorders, including the effects of drugs of abuse, where sleep disturbances are prevalent and often the first symptom of disease or its relapse [37,38,44–47] Remarkably, changes in the two measures that form the SCI, REMS intensity and NREMS duration, are the most prominent in these disorders. A tendency for preserving the SCI may explain why the changes in these two parameters that occur in pathological conditions can be coordinated, no matter which direction the shift is in [37,38,44–53]. For instance, in affective disorders, an increase in REMS intensity is typically accompanied by a decrease in NREMS episode duration, particularly well documented in the first sleep cycle and often referred to as short latency to REMS [37,38,44–52]. These changes are found to correlate with disease severity and treatment outcomes [37,38,44–53] and, for major depression, are widely accepted as a diagnostic biomarker in patients [37,38,44–49] and potential vulnerability marker in family members [50].

In contrast, patients with Parkinson's disease exhibit reduced REMS intensity and an increased duration of the first NREMS episode [53]. Detailed quantitative assessment of REMS intensity over the entire sleep period should determine the extent to which the invariant relationship between NREMS and REMS is preserved or altered in these disorders. It is thus especially important to recognize that the frequently used semi-quantitative analyses of REM density are not suitable for detailed studies on sleep dynamics and the SCI (Figure 7).

### 3.3. State Instability and Strength of Interaction Define the Sleep–Wake Cycle

Figure 11 illustrates the time-dependent behavior of the only independent variable of the wave model, the regulating parameter $x$ and the associated energy of state instability $\varepsilon$, which increase when the system's stability is altered. In physics, the concept of a regulating parameter plays a fundamental role in the study of complex systems. It provides insights into a physical system's behavior, properties, and transitions between phases or states. These regulating parameters can encompass variables like temperature, pressure, or internuclear distance in molecules. Given that the sleep state is a complex physiological phenomenon, the identity of its regulating parameter remains to be elucidated. The sleep state involves intricate reciprocal interactions of various physiological functions, including cognition, immunity, metabolism, and more. At the molecular level, it coordinates essential processes like hormonal secretion, antioxidant defense, and DNA repair. The question of whether a single central biochemical or systemic parameter governs the sleep state or if the regulating parameter reflects the combined impact of all these processes on state stability, integrated at the level of brain nuclei involved in state transitions [1], remains unresolved. For the purpose of the mathematical modeling presented here, the exact nature of this parameter becomes less critical. It can be conceptualized as a simplified representation, a one-dimensional reduction, of numerous regulatory physiological parameters that collectively determine the system's ability to maintain stability. Nevertheless, making precise predictions about its dynamics throughout the wake and sleep periods under normal and disrupted conditions allows the wave model to function as a valuable tool for testing pertinent hypotheses regarding the nature of this regulating parameter.

The homeostatic relationship between sleep and wake states is widely acknowledged. However, prior models did not include specific parameters like the homeostatic setpoint or threshold, relying instead on the constraints imposed by the circadian clock [2,15]. In contrast, the wave model of sleep dynamics formalizes the homeostatic relationship between wake and sleep states by introducing a critical homeostatic energy threshold situated near the equilibrium setpoint where the two potential curves intersect (Figure 2). This threshold marks the region of robust interaction between the wake and sleep states and, as a result, also serves as the site of resonance amplification for the duration of coherent superposition. The position of this homeostatic threshold can be estimated based on the peak of the normal bell-shaped curve of REMS duration. Notably, the peak and thus the threshold might not be reached in case an individual had an insufficiently long sleep period (Figure 10). Variations in this homeostatic threshold among individuals may hold significance for understanding differences in normal sleep patterns and sleep disorders.

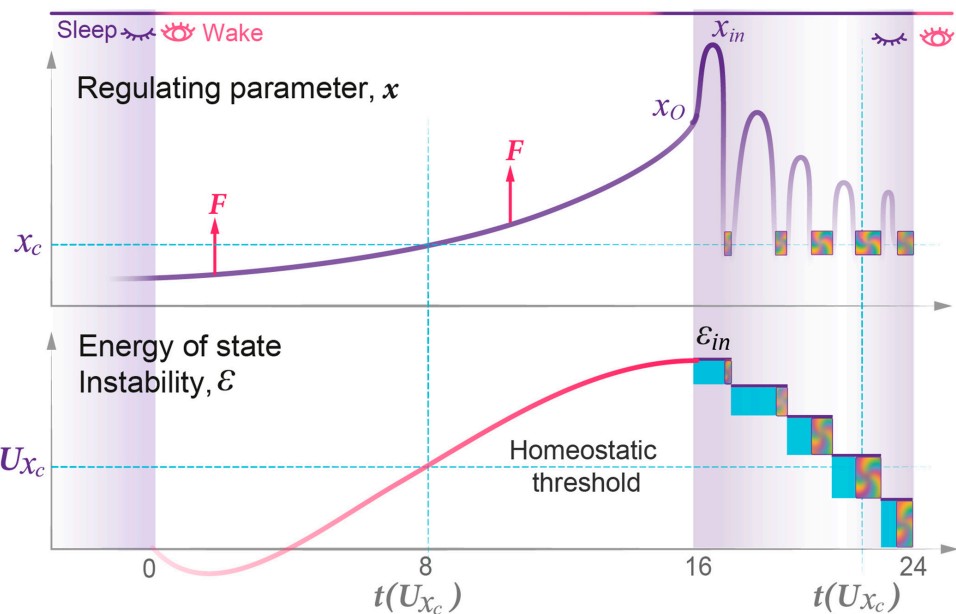

**Figure 11.** Time-dependent changes in the regulating parameter of state stability and energy of state instability. **Top panel**: Time-dependent changes in the regulating parameter of state stability $x$ (vertical axis) over wake (white area) and sleep (purple area). In sleep, the period of $x$ oscillations corresponds to the duration of consecutive NREMS episodes, while a decline in their amplitude squared corresponds to the drop in NREMS intensity. Each cycle ends around $x_c$ (horizontal cyan line), the region of sleep–wake equilibrium where REMS episodes occur (multicolor blocks). $F$—the driving force (red arrow) that increases $x$. **Bottom panel**: Time-dependent increase in the energy of state instability $\varepsilon$ (vertical axis) over the wake state (white area) and stepwise decline over consecutive sleep cycles (block width—duration; NREMS—cyan; REMS—multicolor). Small reduction in $\varepsilon$ at the start of wake period corresponds to sleep inertia. Maximal energy level reached ($\varepsilon_{j_{in}}$) defines initial NREMS intensity. Linearly increasing portions of energy released $\Delta \varepsilon$ correspond to REMS intensity (block height). Horizontal cyan line—potential energy of homeostatic threshold, $U(x_c)$. Vertical cyan line—timing of $U(x_c)$, with daytime timing around mid-afternoon peak in sleep propensity. Horizontal axis—time since awakening (hours).

### 3.4. Future Directions

The wave model has demonstrated a remarkable accuracy in describing normal sleep architecture and dynamics. However, there are obvious additional factors that have to be accounted for while analyzing sleep dynamics and its responses to perturbations, whether intrinsic, environmental, or pharmacological.

It is well-established that both wake and sleep states in humans are regulated not only by a homeostatic mechanism but also by the circadian clock system. Among the four sleep measures analyzed in this study, three are largely independent of circadian regulation, namely NREMS episode duration, intensity, and REM density [36,54]. This aligns with the wave model's premise that the circadian clock modulates the strength of interaction between the sleep and wake states, thus influencing sleep and wake propensities and the duration of REMS episodes [54]. Conversely, the dynamics of the duration and intensity of NREMS and REMS intensity are confined to the sleep state ($U_S$, Figure 2); they do not depend on the strength of the sleep–wake interaction and for this reason are not directly impacted by the circadian clock. While this study focuses on investigating typical sleep patterns under regular entrained conditions, the specific contributions of circadian and homeostatic forces to the sleep–wake interaction will have to be addressed in a separate report.

Another significant concern to be addressed by the wave model is the probability of state transitions, which is reflected in sleep propensity or wake propensity. Empirical evidence of a bimodal daily pattern of sleep propensity is primarily attributed to circadian modulation. Nevertheless, the wave model suggests that the homeostatic interaction between the wake and sleep states may exhibit nonlinearity and also influence the daily profile of sleep propensity.

In the fields of physics and chemistry, the precise mathematical description of a system's normal behavior has proven to be a valuable tool for predicting and comprehending the effects of specific disruptions and discovering strategies to counteract them. The overarching objective of quantitatively modeling sleep dynamics is to enhance our understanding of both normal sleep and sleep-related pathologies and, ultimately, to uncover innovative preventative and therapeutic approaches. Notably, understanding certain pathological alterations in sleep may necessitate a better understanding of wake state architecture and dynamics. The notion of an ultradian structure within the wake state has been a subject of study for decades, dating back to Kleitman's basic rest–activity cycle (BRAC) [55], but has yielded inconclusive results [56]. We anticipate that the wave model could provide valuable insights to help elucidate previous observations or that its predictions might aid in further investigations into the ultradian structure of the wake state.

Finally, we would like to emphasize that the sleep and wake probability waves that the model operates with are not of quantum mechanical nature. It is well established that stochastic classical systems can mimic the dynamics of probability waves observed in quantum physics [25–29]. We consider our findings another example of this phenomenon. We propose that the conceptual and mathematical framework of wave mechanics can be applied to explore various aspects of both wake and sleep states, including their responses to chronic sleep deprivation, circadian phase shifts, disease states, pharmacological interventions, or environmental factors.

## 4. Methods

In this section, we first outline the analogies between the dynamics of sleep and wake states and the dynamics of electronic states in the well-established model of a diatomic molecule that informed our work (a–f). Next, in Methods 1–15, we detail the wave model of normal human sleep dynamics and explain how the comparisons between the experimental data and theoretical predictions were made. Method 16 describes the experimental datasets (ours and of others) that were used to validate the model and includes the Inclusion and Ethics statement for our dataset. Method 17 details the statistical testing.

It is important to note that human sleep is a stochastic process, as reflected in the high inter-individual and night-to-night variation of sleep architecture. Therefore, this work is intended to model average sleep patterns in groups of human subjects studied under well-controlled laboratory conditions. Additionally, it should be noted that the wave model in this paper only addresses normal sleep in adult humans.

### 4.1. Analogies between the Wave Model of Sleep and the Model of a Diatomic Molecule

In our exploration of the dynamics of the sleep state, we view it as an outcome of an interaction of two probability waves, each representing the sleep and wake states. These waves emerge from the intricate workings of numerous biochemical oscillators—homeostatic feedback loops—critical for the organism's structural and functional integrity. As we developed the mathematical framework for dynamic sleep architecture, we drew inspiration from a structural analogy with a model from quantum mechanics: that of a diatomic molecule [24]. Several key parallels between these seemingly disparate domains became evident:

(a) *Fast and slow components*: In a diatomic molecule, the dynamics encompass reciprocal rapid changes in electronic states alongside the slower adjustments of the regulating parameter, the distance between the molecule's two nuclei (R). In our model, we suggest that changes in the biochemical and electrochemical processes that form sleep and wake states are relatively fast, whereas variations in state stability, reflected in the regulating parameter of state stability, denoted as $x$, occur more gradually, akin to alterations in R. In both types of systems, there is a reciprocal relationship between their fast and slow components. As noted in Section 3, the sleep state engages in reciprocal interactions with a multitude of physiological functions and it remains unknown whether a single molecular or systemic parameter central to the sleep state exists, or if the regulating parameter reflects the combined effects of multiple processes on state stability that are integrated by the neuronal structures involved in state transitions [1]. For the purpose of the mathematical modeling of sleep dynamics presented here, the regulating parameter $x$ is regarded as a one-dimensional reduction of numerous regulatory physiological parameters that collectively define the system's capacity to maintain stability.

(b) *State stability of the fast component.* In the molecular system, the electron component can be in a stable (ground) or unstable (excited) state, which are states of different symmetry at a given value of R. Changes in R can result in the swap of symmetry of state, such that a former ground state becomes unstable, while a former excited state becomes stable. Similarly, in our model, the stability of sleep and wake states is determined by the regulating parameter $x$, and changes depend on variations in $x$ value. Low $x$ values favor the wake state and high $x$ values favor the sleep state.

(c) *The interaction and feedback relationship between the fast and slow components.* For different electronic states in a diatomic molecule, the energy of the fast (electronic) component depends on R, and this dependence creates a potential energy U(R) for the slow nuclear motion. We expect a similar relationship within sleep and wake dynamics, where parameter $x$ regulates the stability of the underlying fast processes, e.g., individual homeostatic loops. In turn, those fast processes modulate the dynamics of the regulating parameter $x$, creating distinct potential energies $U_S(x)$ and $U_W(x)$, respectively.

(d) *Probability waves.* The wave nature of probabilistic processes can be illustrated by probability waves, or de Broglie waves, which describe the dynamics of the electronic and nuclear components in a diatomic molecule. The nondeterministic nature of the sleep and wake processes, as well as the coherent dynamics of their slow and fast components, suggests the use of quantum mechanical analogies in the description of sleep architecture.

(e) *State transitions.* The probabilistic transitions between electronic states of different symmetry, that is the swapping of stable and metastable (excited) states, can occur within certain restricted regions of the R parameter where the electronic energies of different states have close values. We predict that transitions between the sleep and wake states will have similar behavior in the region of crossing or pseudo-crossing of the $U_S(x)$ and $U_W(x)$ potential curves. $x_c$ represents the point of sleep–wake homeostatic equilibrium, with $Ux_c$ as the homeostatic energy threshold or setpoint (Figure 2).

(f) *Discrete energy spectra of the stationary probability waves.* In the molecular system, stationary probability waves have a discrete spectrum of energy for both the electronic and nuclear components, the latter being represented by R-vibrations. Analogously, in our model, we introduce the energy parameter $\varepsilon$, which represents the measure of instability for either the wake or sleep state. An increase in state instability leads to an increase in $\varepsilon$.

*4.2. Mathematical Apparatus of the Wave Model of Sleep Dynamics*

4.2.1. Wave Equations for the Sleep and Wake States

It has been demonstrated through both experimental and theoretical studies that classical systems can mimic behaviors commonly associated with quantum mechanics, such as energy level quantization, tunneling, spin structures, and double-slit interference, among others. These effects have been observed in some macroscopic systems containing classical stochastic waves near instability threshold [25–29]. We thus suggested that the probability waves for sleep (S) and wake (W) can be also described using a mathematical analogy with probability waves in quantum mechanics. This enabled the use of the two-component Schrödinger equation for the slow nuclear motion in diatomic molecules to describe the motion of regulating parameter $x$ in our model.

The amplitudes and phases of S and W probability waves are described by the time-dependent wave functions $\Psi_S(x,t) \times |s,x\rangle$ and $\Psi_W(x,t) \times |w,x\rangle$, where $|s,x\rangle$ and $|w,x\rangle$ are the wave functions of the fast intrinsic variables of underlying S and W processes, which are regulated by the $x$ variable. These functions, which are analogues of the wave functions of different electronic states in a diatomic molecule, are orthogonal at any $x$ and $x'$ because of $<x,s|w,x'> = 0$. Functions $\Psi_S(x,t)$ and $\Psi_W(x,t)$ describe the dynamics of variable $x$ in both S and W states. Two-component wave equations can be written in the matrix form:

$$i\frac{\partial}{\partial t}\begin{pmatrix}\Psi_S(x,t)\\\Psi_W(x,t)\end{pmatrix} = \begin{pmatrix}\hat{H}_{SS}(x) & V_{S,W}(x,t)\\V_{W,S}(x,t) & \hat{H}_{WW}(x)\end{pmatrix}\begin{pmatrix}\Psi_S(x,t)\\\Psi_W(x,t)\end{pmatrix} + \hat{D}(x,t)\begin{pmatrix}\Psi_S(x,t)\\\Psi_W(x,t)\end{pmatrix}, \quad (1)$$

$$\hat{H}_{SS}(x) = -\frac{1}{2}\frac{d^2}{dx^2} + U_S(x) \text{ and } \hat{H}_{WW}(x) = -\frac{1}{2}\frac{d^2}{dx^2} + U_w(x), \quad (2)$$

where $U_S(x)$ and $U_W(x)$ are the potential energies modulating the propagation of S and W waves, respectively, and $V_{S,W}(x,t) = V_{W,S}^*(x,t)$ is the matrix element of the operator responsible for the interaction between S and W states. Note that $V_{S,W}$ reflects both the $x$-dependent and time-dependent interstate interaction, including the entrained 24 h periodicity in the sleep–wake cycle that is controlled by the circadian system [54]. Efficient S↔W state transitions can be induced by a S–W interaction near the crossing point of the $U_S(x)$ and $U_W(x)$ potential curves. Although the potentials $U_S$ and $U_W$ exist simultaneously at the same coordinate $x$, each affects only the corresponding wave. The time, coordinate, and energies in Equations (1) and (2) are dimensionless values measured in their specific units. The normalization of the wave functions $\int_{-\infty}^{+\infty} dx\left(|\Psi_S|_2 + |\Psi_W|^2\right) = 1$ provides the probabilities of the realization of the S or W state: $P_S(t) = \int_{-\infty}^{+\infty} dx\left|\Psi_S\right|^2$ and $P_W(t) = \int_{-\infty}^{+\infty} dx\left|\Psi_W\right|^2$.

Operator $\hat{D}(x,t)$ in Equation (1) includes driving force $F$, which increases the energy of the system, and forces that damp the kinetic energy of the $x$ coordinate. The closest quantum analogy with Equations (1) and (2) is the system of equations for the nuclear motion of diatomic molecules with two electronic states of different symmetry [24,57]. In this mathematical analogy, electronic states represent fast underlying processes regulated by the value of microscopic interatomic distance. However, in the dynamics of macroscopic probability waves [25–29], which we assume S and W waves are, the time and spatial scales are many orders of magnitude greater than those described in quantum physics. The time evolution of wave functions $\Psi_S(x,t)$ and $\Psi_W(x,t)$ determines the entire dynamics of S and W states, and consequently of sleep architecture.

### 4.2.2. S and W Stationary Waves

Complete sets of eigen functions (stationary waves) can be used as a standard mathematical tool for the determination of wave functions satisfying Equations (1) and (2). The equations for S and W stationary waves can be obtained by neglecting the S–W interaction ($V_{S,W} = 0$) and the action of the $\hat{D}(x,t)$ operator. In this case, the stationary S waves $\Psi_{S,j}(x,t) = \exp(-i\,\varepsilon_j\,t) \times \varphi_j(x)$ and W-waves $\Psi_{W,k}(x,t) = \exp(-i\,\varepsilon_{w,k}\,t) \times \psi_k(x)$ are expressed via two sets of independent eigen functions $\varphi_j(x)$ and $\psi_k(x)$ and their eigen energies $\varepsilon_j$ and $\varepsilon_{w,k}$, respectively. Since the objective of our model is an accurate description of sleep architecture, we mainly consider the stationary S waves representing the bound states of the potential $U_S(x)$ with a discrete set of energies $\varepsilon_j$ ($j = 0, 1, 2, 3 \ldots$). Experimental data on the architecture of the W state are limited, and an analysis of W stationary waves is beyond the scope of our model. However, the wave model suggests that the interaction of S and W states plays a key role in energy relaxation during sleep, specifically in the process of energy release, and that this is reflected in sleep architecture.

### 4.2.3. Energy Spectra of the S waves in the Morse Potential

The quasi-periodic nature of sleep cycles with gradually reducing durations of NREMS episodes $T_{NR}$ (Figure 1b) indicates that neither parabolic nor rectangular potential wells can describe the sleep process (Supplemental Figure S2). In contrast, the Morse potential $U_S(x)$, commonly used in atomic and molecular physics, provides a set of energy levels $\varepsilon_j$ that are sufficient for an accurate quantitative description of S wave propagation and the energy relaxation process. The one-dimensional Morse potential is given by the equation:

$$U_S(x) = U_0 \times \left( e^{-2\frac{x}{\sigma}} - 2e^{-\frac{x}{\sigma}} \right), \tag{3}$$

where $U_0$ and $\sigma$ are positive constants that describe the depth and width of the potential well, respectively. Our model is designed to predict the relative values of $T_{NR}$ in different sleep cycles, so we can scale the depth of $U_S$ potential, $U_0 = 1$. As a result, energies $\varepsilon_j$ at each level and the corresponding eigen wave functions depend on the single parameter $\sigma$. The spectra of discrete energy levels $\varepsilon_j$ of the Morse potential are given by the following equation [24]:

$$\varepsilon_j = \varepsilon_j(\sigma) = -\left( 1 - \frac{j + \frac{1}{2}}{N(\sigma)} \right)^2 \quad (j = 0, 1, 2, \ldots j_{Max}) \tag{4}$$

where $N(\sigma) = \sqrt{2}\sigma$ is the dimensionless parameter that regulates the total number of discrete energy levels in the Morse potential, $j_{Max} = \left[ N(\sigma) - \frac{1}{2} \right]$.

The discrete energy spectrum and wave functions of stationary waves can also be approximated using the quasi-classical Bohr–Sommerfeld model [24,58]. The quasi-classical approximation illustrates the connection between wave and particle dynamics, and describes the propagation of the wavepackets of probability waves. We will later consider the dynamics of the quasi-classical wavepacket of S waves and thus determine the NREMS episode durations, $T_{NR}$. In the Morse potential, the period of oscillations of the wavepacket moving with an energy close to $\varepsilon_j$ declines as energy $\varepsilon_j$ drops, while energy gaps $\Delta\varepsilon(j) = \varepsilon_j - \varepsilon_{j-1}$ between consecutive levels increase (Figure 2; Supplemental Figure S2).

### 4.2.4. Energy Relaxation and Structure of the S Wavepacket

A spontaneous W $\rightarrow$ S transition initiating sleep generates a wavepacket of S waves with initial energy $\varepsilon \approx \varepsilon_{j_{in}}$. Relaxation of instability energy $\varepsilon$ in stepwise transitions, $j \rightarrow j - 1$, occurs through the release of discrete portions of energy $\Delta\varepsilon(j) = \varepsilon_j - \varepsilon_{j-1}$. The level $j(n)$ occupied by the wavepacket during the n-th sleep cycle of the relaxation process is determined by the initial level $j_{in}$ and the order number $n$ of the sleep cycle (Figures 1 and 2), $j(n) = j_{in} - n + 1$. In the general case, the exact composition of the wavepacket with an

arbitrary energy $\varepsilon$ includes all stationary S waves belonging to the discrete and continuous spectra. At the same time, in the quasi-classical approximation, it can be mostly represented by a single S wave from $j$ energy level, if $\varepsilon$ is close to the energy level $\varepsilon_j$. For simplicity, we neglect the wavepacket dispersion over the entire relaxation process and consider that the wavepacket energy $\varepsilon_{j(n)}$ is reduced only in the stepwise $j(n) \rightarrow j(n+1)$ transitions during REMS episodes.

### 4.2.5. Quasi-Classical Motion of the Wavepacket

At a large $j$, the period $T_j$ of quasi-classical oscillations of the center of the wavepacket corresponds to the period of motion of a classical particle with energy $\varepsilon_j$ bouncing between the potential walls of $U_S(x)$. $T_j$ can be expressed through the energy difference between the neighboring energy levels [57]: $T_j = 2\pi / \left( \frac{\partial \varepsilon_j}{\partial j} \right) \approx 2\pi / \Delta \varepsilon(j)$, where the derivative $\frac{\partial \varepsilon_j}{\partial j}$ has been replaced with the energy gap $\Delta \varepsilon(j)$. The wavepacket motion is also quasi-classical at lower energy levels $j$ where the Morse potential is similar to the potential of the harmonic oscillator (Equation (3) and Figure 2). The unitless value of $T_j$ is given by the equation:

$$T_j = \frac{2\pi}{\Delta \varepsilon(j)} = 2\pi \frac{1}{\varepsilon_j(\sigma) - \varepsilon_{j-1}(\sigma)} = 2\pi \frac{\sigma^2}{\sqrt{2}\,\sigma - j} \tag{5}$$

The discrete set of $T_j$ reflects the discrete structure of the energy spectra $\varepsilon_j$ of S stationary waves. The set of consecutive $T_{j(n)}$ periods with $j = j(n)$ predicts the relative durations of consecutive NREMS episodes $T_{NR}(n)$, i.e., periods of quasi-classical motion in the Morse potential. These values depend only on the width of potential well $\sigma$ and actual level $j$ occupied by the wavepacket during the $n$-th sleep cycle.

### 4.2.6. Interaction between S and W Stationary Waves and Their Coherent Superposition

The independence of S and W states is strongly violated within the $\delta x_c$ region of nonadiabatic behavior around crossing point $x_c$ (Figures 2, 5a and 8), where the interaction $|V_{S,W}(x,t)|$ is larger than or comparable to the difference between the potential energies: $|U_S(x) - U_W(x)| \lesssim |V_{S,W}(x,t)|$ [23]. Thus, inside $\delta x_c \sim |V_{S,W}(x,t)| / |\frac{\partial U_S}{\partial x} - \frac{\partial U_W}{\partial x}|_{x=x_c}$, stationary S and W waves can form a coherent superposition or entanglement. The formation of S–W entanglement inside the moving wavepacket can be clarified by a simplified example in which an entangled state is constructed from isolated stationary S and W waves. A two-state model includes a single S wave eigenstate $\varphi_s(x)$ and single eigenstate of W-wave $\psi_w(x)$ with energies $\varepsilon_s$ and $\varepsilon_w$, and a time-independent potential of S–W interaction $V_{S,W}(x,t) = V(x)$ can be used to explain this phenomenon. This interaction creates new stationary waves with new energies $E_{a,b}$ and wave functions $\Psi_{a,b}(x,t) = \chi_{a,b}(x) \times e^{-i\,E_{a,b}t}$, where new eigen functions $\chi_{a,b}(x)$ are expressed via the coherent superposition of unperturbed $\varphi_s(x)$ and $\psi_w(x)$ waves of S and W states:

$$\begin{cases} \chi_a(x) = \cos \frac{\theta}{2} \times \varphi_s(x) + \sin \frac{\theta}{2} \times \psi_w(x)\,, \\ \chi_b(x) = -\sin \frac{\theta}{2} \times \varphi_s(x) + \cos \frac{\theta}{2} \times \psi_w(x)\,, \end{cases} \tag{6}$$

The eigen energies $E_{a,b}$ and coefficients of the coherent superposition $\cos \frac{\theta}{2}$ and $\sin \frac{\theta}{2}$ are given by the equations:

$$E_{a,b} = \frac{\varepsilon_s + \varepsilon_w}{2} \mp \sqrt{\frac{\Delta \varepsilon_{sw}{}^2}{4} + \widetilde{V}_{s,w}{}^2}\,, \quad \Delta \varepsilon_{sw} = \varepsilon_s - \varepsilon_w\,, \text{ and } \cos \theta = \frac{1}{\sqrt{1 + \left( 2\frac{\widetilde{V}_{s,w}}{\Delta \varepsilon_{sw}} \right)^2}}\,, \tag{7}$$

where $\widetilde{V}_{SW} = \int_{-\infty}^{+\infty} dx\, \varphi_s^*(x) V(x) \psi_w(x)$ is the matrix element of the S–W interaction potential and $\theta$ is the mixing angle. For simplicity, we assumed zero values of the diagonal matrix elements, $\widetilde{V}_{SS} = \widetilde{V}_{WW} = 0$. Note that the energies $E_{a,b}$ of the new states differ from the energy levels of $U_S$ and $U_W$ potentials. Probabilities $p_s$ and $p_w$ to detect the characteristic features of S and W states in new stationary wave $\Psi_a(x,t)$ are expressed in

terms of the coefficients of coherent superposition: $p_s = \cos^2 \frac{\theta}{2}$ and $p_w = \sin^2 \frac{\theta}{2}$. Under the condition of weak state interaction, i.e., $\theta \ll 1$, the new stationary wave $\chi_a(x)$ is mainly represented by $\varphi_s(x)$, occupying primarily the region of S stability $x > x_c$. The second stationary wave $\chi_b(x)$ is located predominantly within the region of W stability $x < x_c$ and it is represented mostly by $\psi_w(x)$. The parameter of perturbation $\beta = \left| \frac{\widetilde{V}_{s,w}}{\Delta\varepsilon_{sw}} \right|$ describes the relative strength of interaction $\widetilde{V}_{s,w}$ and regulates the fractions of W and S states within their coherent superposition, $\Psi_{a,b}(x,t)$. In the case of $\beta \to 0$, i.e., weak S–W interaction, or very large energy difference $\Delta\varepsilon_{sw}$, the entanglement is not formed. In contrast, a large value of $\widetilde{V}_{s,w}$ or a small difference between S and W energy levels ($\beta \gg 1$ or $\Delta\varepsilon_{sw} \to 0$) creates the conditions for strong coherent mixing of S and W waves. In this case, the mixture angle is $\theta = \frac{\pi}{2}$, the wave mixture coefficients are $\cos \frac{\theta}{2} = \sin \frac{\theta}{2} = 1/\sqrt{2}$, and the coherent S–W superpositions $\frac{1}{\sqrt{2}}[\varphi_s(x) \pm \psi_w(x)]$ include equal fractions of S and W waves.

The condition of $\Delta\varepsilon_{sw} \to 0$ can be seen as a resonance between two stationary waves of different symmetry. Accidental resonance can occur between stationary waves contained within different potential wells [59]. Under conditions of wavepacket propagation, its motion is considered as an adiabatically slow process and the mechanism of formation of coherent states includes a similar mixing of stationary waves, with one important difference: the entanglement of S and W states is temporary and lasts for as long as the center of the wavepacket remains inside nonadiabatic region $\delta x_c$.

### 4.2.7. The Coherent Superposition of S and W Waves and Wavepacket Dynamics

In the energy relaxation process, the wavepacket is composed mainly of $\varphi_{j(n)}(x)$ waves and its center moves like a classical particle, bouncing between $U_S(x)$ walls (Figure 2). Within this semi-classical approximation, transitions between W and S states can occur when the center of the wavepacket approaches the $\delta x_c$ region where the crossing of $U_S(x)$ and $U_W(x)$ potential curves enhances the efficiency of the S and W interaction. Within $\delta x_c$, the wavepacket can temporarily incorporate a substantial fraction of stationary W waves, as shown in Equations (6) and (7). The S–W interaction creates a delay $\tau$ in wavepacket propagation and leads to a new temporary state represented by the coherent superposition of S and W waves. Thus, an organism can simultaneously exist in both S and W states over time interval $\tau$, until the wavepacket leaves the $\delta x_c$ region. In our model, this new state corresponds to REMS, coherently incorporating S and W features. The $\tau$ defines the lifetime of the coherent superposition within $\delta x_c$ and thus REMS episode duration $T_{REM}$.

### 4.2.8. Short Time-Delay Induced in Landau–Zener Transition

The presence of the region of enhanced S–W interaction $V_{S,W}$, i.e., the region of nonadiabatic behavior of the S state, leads to additional accumulation of phase $\eta(\varepsilon)$ of the stationary wave with energy $\varepsilon$, and thus to a delay in wavepacket propagation. The resulting time delay $\tau(\varepsilon)$ can be calculated using the energy dependence of the wave phase shift, $\eta(\varepsilon)$: $\tau(\varepsilon) = 2\frac{d\eta(\varepsilon)}{d\varepsilon}$ [60]. The time delay $\tau_{LZ}(\varepsilon)$ of the wavepacket passing through $\delta x_c$ can be also estimated using the semi-classical Landau–Zener model [24]:

$$\delta x_c \sim \frac{|V_{SW}(x_c)|}{|\frac{\partial U_S}{\partial x} - \frac{\partial U_W}{\partial x}|_{x=x_c}} \quad \text{and} \quad \tau_{LZ}(\varepsilon_j) \sim \frac{\delta x_c}{v_j} = \frac{|V_{SW}(x_c)|}{v_j |\frac{\partial U_S}{\partial x} - \frac{\partial U_W}{\partial x}|_{x=x_c}},$$

where $v_j$ is the velocity of the classical particle moving with energy $\varepsilon_j$ through the $x_c$ region. Usually, the Landau–Zener model cannot provide long $\tau$ because interaction potential $V_{SW}(x)$ is considered a small perturbation with respect to $U_S(x)$ and $U_W(x)$. In our model, $\tau_{LZ}$ is the background time delay in the absence of the resonance process. General mathematical analysis and experimental data on the nonadiabatic transitions and phase accumulation for the extended Landau–Zener model have been reported in [61–64].

### 4.2.9. Resonance Enhancement of REMS Episode Duration and Energy Release

Within $\delta x_c$, the presence of the W-wave component of the wavepacket leads to the action of the damping forces included in operator $\hat{D}(x,t)$, as shown in Equation (1). This allows for the wavepacket to release a fraction of its energy $\Delta\varepsilon(j)$, corresponding to the energy gap between the $j$ and $j-1$ levels. At a constant value of damping forces, the efficiency of energy release depends on the strength of the S–W interaction and the time delay $\tau(\varepsilon_j)$. The value of released energy $\Delta\varepsilon(j)$ increases with the number of cycles $n$. Facilitating energy release requires more time for the action of the damping forces, i.e., increasing $\tau(\varepsilon_j)$. This can be accomplished through a resonance condition for the incoming S wave: if the energy of resonance level $\varepsilon_{jc}$ is close to $U(x_c)$, the potential energies of S and W states near their crossing point $x_c$, where the strength of the S–W interaction is high. The temporary capture of the wavepacket into the resonance state augments the lifetime of coherent S–W superposition and thus $T_{REM}$.

### 4.2.10. Mechanisms of Resonance Formation

The formation of a resonance state can occur through different mechanisms. One example is Feshbach resonance [65], in which energy levels of $U_S(x)$ and $U_W(x)$ are located near $U(x_c)$. In this case, $\tau(\varepsilon_j)$ determines the effective time in which the W state is present within the composition of the S wavepacket when the center of the wavepacket is located within the $\delta x_c$ region. Another mechanism by which S wave resonance may occur is through a very strong local interaction between S and W waves, which creates a quasi-bound state localized near $x_c$. In this case, the avoided crossing structure of the adiabatic potentials can support the resonance (quasi-bound) state for incoming S waves and temporarily localize the wavepacket within $\delta x_c$. Detailed mathematical descriptions of these resonance effects have been developed in the theory of slow atomic collisions [66]. For either mechanism, the energy of S wave resonance $\varepsilon_{jc}$ can be formally associated with resonance level $j_c$.

### 4.2.11. Lorentz Resonance Curve

Independent of the mechanism of the resonance, at any energy $\varepsilon_j$ of the S wavepacket, the resonance time delay $\tau(\varepsilon_j)$ is sensitive to the energy difference between $\varepsilon_j$ and resonance energy $\varepsilon_{jc}$. The time delay can be calculated from additional resonance phase $\eta(\varepsilon_j)$ [57] with a simple expression for the resonance value of $\tau(\varepsilon_j)$ given by the resonance Lorentz formula [60,65]:

$$\tau(\varepsilon_j) = 2\frac{d\eta(\varepsilon)}{d\varepsilon}|_{\varepsilon=\varepsilon_j} = \frac{1}{\pi}\frac{\frac{\Gamma}{2}}{(\varepsilon_j - \varepsilon_{jc})^2 + \left(\frac{\Gamma}{2}\right)^2}, \tag{8}$$

where $\varepsilon_{jc}$ and $\Gamma$ are the energy level and width of S wave resonance, respectively. The $j_C$ value can be expressed via $U(x_c)$: $j_c \cong \sqrt{2}\,\sigma\left(1 - \sqrt{|U(x_c)|}\right) - \frac{1}{2}$. It is not an integer because, in the general case, resonance energy does not exactly match $\varepsilon_j$ levels. The dynamic changes in $\tau\left(\varepsilon_{j(n)}\right)$ over the process of stepwise energy relaxation describe relative durations of consecutive REMS episodes $T_{REM}(n)$ over the course of sleep.

### 4.2.12. NREMS and REMS Episode Durations in Absolute Time Units

$T_{NR}(n)$ and $T_{REM}(n)$ in absolute time units (min) can be obtained using Equations (5) and (8) for relative durations $T_{j(n)}$ and $\tau\left(\varepsilon_{j(n)}\right)$ with $j(n) = j_{in} - n + 1$, and the absolute time scaling constants $A_{NR}$ and $A_{REM}$, respectively:

$$T_{NR}(n) = A_{NR}\, T_{j\{n\}} = A_{NR}\, 2\pi \frac{\sigma^2}{\sqrt{2}\,\sigma - j_{in} + n - 1}\,, \tag{9}$$

$$T_{REM}(n) = A_{REM}\, \tau\left(\varepsilon_{J(n)}\right) = A_{REM}\, \frac{1}{\pi}\, \frac{\frac{\Gamma}{2}}{\left(\varepsilon_{j_{in}-n+1} - \varepsilon_{jc}\right)^2 + \left(\frac{\Gamma}{2}\right)^2}\,, \tag{10}$$

To calculate the relative values of $T_{NR}(n)$, two model parameters are needed: the width of $U_S$ potential $\sigma$ and initial level $j_{in}$. To calculate the relative values of $T_{REM}(n)$, two additional model parameters are required: the energy $\varepsilon_{jc}$ of resonance level and resonance width $\Gamma$ (Figure 8). These parameters have been inferred from comparisons of the experimental data on $T_{NR}(n)$ and $T_{REM}(n)$ and the theoretical predictions of these values given by the analytical formulas in Equations (9) and (10) (Supplementary Table S1). Since the model predicts relative times, the time-scaling constants $A_{NR}$ and $A_{REM}$ were used to convert values of relative durations into absolute time units (min).

### 4.2.13. Predictions of NREMS Intensity as a Function of Initial Energy $\varepsilon_{j_{in}}$ and Amplitude of $X$-Oscillations

NREM intensity $I_{NR}(n)$, i.e., the duration of slow-wave sleep (SWS or slow-wave activity), is maximal in the first NREMS episode ($n = 1$) and rapidly decreases over consecutive sleep cycles. This behavior correlates positively with the intensity of $x$-oscillations (Figure 4), presumably because the relative intensities of some processes underlying SWS are regulated by $x$ and may follow similar oscillatory dynamics. At each level $j = j(n)$ occupied by the wavepacket in the energy relaxation process, we expect that $I_{NR}(n)$ is proportional to the intensity of $x$-oscillations, i.e., the square of $x$-amplitudes $L^2(\varepsilon_J)$: $I_{NR}(n) = \kappa(\varepsilon_{j(n)}) \times L^2\left(\varepsilon_{J(n)}\right)$, where coefficient $\kappa\left(\varepsilon_{j(n)}\right)$ regulates the efficiency of SWS generation over the entire sleep process. The $k$-coefficient should be maximal at the upper $j$ level and decline at lower levels: $\kappa\left(\varepsilon_j\right) = const/\left|\varepsilon_j\right|$. Note that the energies of the bound waves are negative $-1 < \varepsilon_j < 0$ and the absolute value $\left|\varepsilon_{j(n)}\right|$ increases as the level index $j(n) = j_{in} + 1 - n$ decreases in the energy relaxation process.

The Morse potential is asymmetric and the measure of the amplitude of nonharmonic $x$-oscillations can be the distance $L(\varepsilon)$ between the right and left classical turning points (Figure 2). For the wavepacket bouncing between the walls of the Morse potential, oscillation amplitude $L(\varepsilon)$ is expressed through the width of potential well $\sigma$ and wavepacket energy $\varepsilon$:

$$L(\varepsilon) = 2\sigma\, \ln\left[\frac{1 + \sqrt{1 - |\varepsilon|}}{\sqrt{|\varepsilon|}}\right], \tag{11}$$

where the value of $\varepsilon$ in the $n$-th sleep cycle is given by $\varepsilon_{j(n)}$ in Equation (4), with $j = j(n)$. Theoretical values of $I_{NR}(n)$ can be normalized to the first cycle and results are given by the formula:

$$\frac{I_{NR}(n)}{I_{NR}(1)} = \frac{|\varepsilon_{j_{in}}|}{|\varepsilon_{j(n)}|} \left(\frac{L\left(\varepsilon_{j(n)}\right)}{L\left(\varepsilon_{j_{in}}\right)}\right)^2, \tag{12}$$

were wavepacket energies $\varepsilon_{j(n)}$ and oscillation amplitudes $L\left(\varepsilon_{j(n)}\right)$ are given by Equations (4) and (11) with $j = j(n)$. In Figure 4b, theoretical predictions of relative NREMS intensities for different sleep cycles are compared to experimental data obtained from independent groups of healthy volunteers studied under regular sleep conditions and

extended sleep conditions (Method 16, below). It is important to note that the theoretical predictions for $I_{NR}(n)$ in Equation (12) do not include any adjustable parameters, and all necessary model parameters were inferred from the observed durations of REMS and NREMS episodes (Supplementary Table S1).

### 4.2.14. Dynamics of REMS Intensity Depends on Energy Release

According to the model, a rise in REMS intensity $I_R(n)$ with $n$ reflects an increase in the portion of energy $\Delta\varepsilon(j(n))$ released within the $n$-th REMS episode, $I_R(n) \propto \Delta\varepsilon(j(n))$. Theoretical values for $I_R(n)$, scaled to REM intensity in the first REMS episode ($n = 1$), show linear dependence on cycle order number $n$:

$$\frac{I_R(n)}{I_R(1)} = \frac{\Delta\varepsilon(j_{in} + 1 - n)}{\Delta\varepsilon(j_{in})} = 1 + \frac{n-1}{\sqrt{2}\,\sigma - j_{in}} \tag{13}$$

Figure 5b illustrates the theoretical linear dependence of REM intensity on $n$, calculated using Equation (13) with the values of $\sigma$ and $j_{in}$ determined in the analysis of REMS and NREMS episode durations for the regular sleep opportunity group (Supplementary Table S1, Method 16). The slope of the theoretical curve $1/\left(\sqrt{2}\,\sigma - j_{in}\right)$ depends on two model parameters through the single combination $\left(\sqrt{2}\,\sigma - j_{in}\right)$ that indicates an approximate number of energy levels above $j_{in}$. The slope of $I_R(n)/I_R(1)$ increases as the initial level $j_{in}$ approaches the maximal value $j_{Max} = \left[\sqrt{2}\sigma - \frac{1}{2}\right]$. Our theoretical predictions for relative REM intensities were computed without any fitting parameters and are in excellent agreement with experimental data collected by us and others [34,35] in three independent groups of young healthy subjects, as shown in Figure 5b.

### 4.2.15. Sleep Cycle Invariant

The wave model predicts the existence of a value that remains constant over consecutive sleep cycles, the sleep cycle invariant (SCI). The SCI is defined as the product of NREMS episode duration, $T_{j(n)}$, and REMS episode intensity, $I_R(n)$:

$$T_{j(n)} \times I_R(n) = \frac{2\pi}{\Delta\varepsilon(j(n))} \times const \times \Delta\varepsilon(j(n)) = 2\pi \times const. \tag{14}$$

The SCI value does not depend on the value of energy gap, $\Delta\varepsilon(j)$, and can serve as a signature of sleep cycle integrity or overall sleep quality. Note that, while calculating the SCI for the first sleep cycle, the shorter duration of $T_{j(n=1)}$ (due to $x_o$ position; Figure 2) has to be accounted for and the experimental value of $T_{j(n=1)}$ has to be multiplied by 4/3.

### 4.2.16. Datasets

The model was validated against several sets of polysomnographic data collected under well-controlled sleep laboratory conditions in groups of young healthy subjects with normal sleep patterns. Those presented in Figures 3–5, 7 and 9 are described below:

(a) *Regular sleep opportunity protocol.* The representative group data for regular sleep shown in Figures 3–5, 7 and 9 were part of our larger study on the circadian regulation of sleep and hormonal functions ("Multimodal Circadian Rhythm Evaluation" PI: IVZ), which will be reported in full elsewhere. The study was conducted in accordance with the Declaration of Helsinki on Ethical Principles for Medical Research Involving Human Subjects, adopted by the General Assembly of the World Medical Association, and approved by the Boston University Institutional Review Board. All the participants provided written informed consent. The subjects were 24 young healthy male volunteers (Mean ± SEM: 24.5 ± 4.4 years of age, ranging 19–34 years of age) who were selected based on the following self-reported criteria: 7–9 h of habitual nighttime sleep, small (<1.5 h) changes in sleep length on weekends, no sleep complaints, no history of chronic disorders or regular

medications, no recent trans-meridian travel, no drug use, no smoking, habitual coffee consumption not exceeding 3 cups a day.

Over the two weeks prior to the inpatient part of the study, participants' sleep–wake cycles were documented using activity monitors (Phillips Inc., Andover, MA) and a sleep log. Starting on Friday night, subjects spent 3 consecutive nights in the General Clinical Research Center of Boston University School of Medicine. Time in bed was scheduled individually to correspond to the habitual bedtime and subjects were allowed to stay in bed for 9 consecutive hours. Sleep was recorded using polysomnography (Nihon Kohden PSG system), as per standard techniques, and sleep stages were visually scored for consecutive 30 s epochs [67]. To be included in the regular sleep data set, individual sleep nights had to satisfy the following criteria: sleep efficiency of not less than 85% and the absence of sleep apnea or other symptoms of sleep disorders (n = 39 nights total). NREMS–REMS cycles were defined by the succession of an NREMS episode of at least 10 min duration and a REMS episode of at least 3 min duration. No minimum criterion for REMS duration was applied for the completion of the last cycle. An NREMS episode was defined as the time interval between the first two epochs of stage 2 and the first occurrence of REMS within a cycle. A REMS episode was defined as the time interval between two consecutive NREMS episodes or as the interval between the last NREMS episode and the final awakening.

(b) *The extended sleep opportunity protocol* is described in detail in the original reports by Barbato and Wehr [30] (n = 11 subjects, 308 nights total) and Barbato et al. [31] (n = 8 subjects, 208 nights total). In brief, the studies were conducted on healthy male volunteers, 20–34 years of age, who were studied for 4 weeks, with regular activities over 10 h of light and bedrest over 14 h of darkness, when they were encouraged to sleep. The data used in the present study (Figures 3–5, 7 and 9) were obtained from Tables 1–3 of [30] and Table 2 of [31], in consultation with Dr. Barbato.

(c) Slow-wave activity (SWA) data presented in Figure 4c were obtained from Table 1 of the original report by Dijk et al. [32] (9 male subjects, ages 22–26 year). Two baseline nights were used in the analysis (n = 18 nights) and one recovery night per subject was documented (n = 9 nights) following 36 h sleep deprivation.

(d) REMS intensity (REM density) data presented in Figure 3e were obtained from original reports by Aserinsky [34] (11 normal subjects, young males and females) and Marzano et al. [35] (50 normal subjects, young males and females). In both studies, the subjects were identified as university students. REM density data per sleep cycle of baseline night recordings were obtained from p. 550 of [34] and Figure 1 of [35].

### 4.2.17. Statistical Testing

Goodness of fit was assessed using a one-sided $\chi^2$ test, with the degrees of freedom equal to $n - 1 - p$, where $n$ is the number of independent sleep cycle values being fit, and $p$ is the number of model parameters being fit ($p = 4$ for episode duration fits, and $p = 0$ for intensity fits). Note that due to the normalization employed, the number of independent sleep cycle values in each plot is one less than the number of cycles shown. $\chi^2$ and $R^2$ calculations were carried out using standard R functions.

### 5. Conclusions

- The wave model of sleep dynamics represents the sleep and wake states as interacting probability waves. By applying the mathematical apparatus of wave mechanics developed for the analysis of probability waves and state transitions on the molecular level, the model provides a precise quantitative description of the typical dynamics of four principal measures of normal sleep architecture, including the durations and intensities of consecutive episodes of NREM and REM sleep, as documented in experimental groups of young healthy adults.
- The model demonstrates that the duration and intensity of consecutive episodes of NREM sleep, as well as the intensity of REM sleep episodes, only reflect the behavior of the sleep probability wave. The variability in these sleep measures between

experimental groups can be described by two model parameters: first, the width of the potential well containing the sleep wave, which depends on the habitual sleep duration; and second, the energy level of state instability reached by the time of sleep initiation, which depends on the duration of prior wakefulness.

- The model describes REM sleep as a coherent superposition of sleep and wake waves. Accordingly, an accurate description of the duration of REM sleep episodes depends on the strength of interaction between these two waves, which is maximal in the region of their homeostatic equilibrium and further enhanced by the resonance between them.

- The model establishes an invariant relationship between NREM and REM sleep by predicting that the product of NREM sleep duration and REM sleep intensity should normally remain constant over consecutive sleep cycles. The analysis of experimental group data collected in young adults with high quality sleep confirms this prediction, indicating an intrinsic connection between NREM and REM sleep as part of the same two-step sleep process.

**Supplementary Materials:** The following supporting information can be downloaded at: https://www.mdpi.com/article/10.3390/clockssleep5040046/s1, Figure S1: Consecutive periods of energy relaxation in potential wells of different shape. Figure S2: Actual position of the initial and resonance levels within the Us potential well. Table S1: The model parameters of normal sleep, regular and extended.

**Author Contributions:** V.K. and I.V.Z. designed the research; I.V.Z. performed the sleep research; V.K. conducted the mathematical modeling; V.K. and I.V.Z. wrote the paper. All authors have read and agreed to the published version of the manuscript.

**Funding:** This work on the mathematical modeling of sleep process was supported by the Chaikin-Wile Foundation (UConn, AG171189 PI: V.K). Experimental sleep research was supported by Pfizer Inc., Cambridge, MA, USA (BU, 55202665, PI: I.V.Z.). We thank Peter Kharchenko for the constructive discussions and valuable advice, Olga Kharchenko for the illustrations, and our Boston University team for technical assistance in data collection and analysis.

**Institutional Review Board Statement:** The study was conducted in accordance with the Declaration of Helsinki, and approved by the Institutional Review Board of Boston University School of Medicine (protocol code H-33035 and date of approval is 15 November 2015).

**Informed Consent Statement:** Informed consent was obtained from all subjects involved in the study.

**Data Availability Statement:** Correspondence and requests for materials should be addressed to vkharchenko@cfa.harvard.edu and irina.zhdanova@bio-chron.com. The dataset generated by the co-author (I.V.Z.) and analyzed in the current study (Method 16, Regular sleep) is available from the corresponding author on reasonable request. The mathematical algorithms of the wave model of sleep dynamics in "Wolfram Mathematica" format are available from the corresponding author on reasonable request.

**Conflicts of Interest:** Author I.V.Z. is employed by the company BioChron LLC. The remaining authors declare that the research was conducted in the absence of any commercial or financial relationships that could be construed as a potential conflict of interest.

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
