# Peer review of "The Wave Model of Sleep Dynamics and an Invariant Relationship between NonREM and REM Sleep"

_2624-5175, doi:10.3390/clockssleep5040046_

Round 1

Reviewer 1 Report

Comments and Suggestions for Authors

This is an interesting paper which provides a stimulating model for the  ultradian rhythm of NREM and REM sleep within the sleep period. The authors suggest a comprehensive wave model which seems to regulate reciprocal occurrence of NREM and REM periods as well as their intensity, although the model appears valid and original, the authors should improve the part which test the model with published data.

As suggested by the authors, NREM intensity is normally evaluated using SWS or SWA. It would not be difficult to find published data on SWA, which is more sensitive than crude SWS, to support their model, either Feinberg or Borbely and more recently Dijk, Aeschbach, have published several papers, with avalaible data, on this topic.   

As well as in their test for the "sleep cycle invariant", the authors use what they call an "hybrid analysis", using data for the two variables from two different studies. They justify this approach to be related to the lack of data on reliable REM intensity, since REM density is mainly measured as a score or number of intervals. Probably the authors should further stress this methodological point and better clarify the meaning of REM intensity. Also since the two variables, NREM an REM intensity show opposite pattern across the night related to progressive decrease of sleep pressure/progressive increase of arousal, is it possible that the SCI simply reflect these processes? Please discuss this point.

It would also helpful to add evidence from the literature which can support the model prediction.

Minor issues

Please clarify the meaning  of  " abundance" , "sleep abundance"

In several part of the paper (i.e. lines 193, 206, 343, 439.... ) there is a systematic typo, citing Wehr (i.e. Wehrmacht! Wher, etc.), please check.

lines (1003-1008), it seems that the two studies cited (30) and (31) have different number of subjects and sleep records, as you report in figure 7 , please check and report correctly.

Reviewer 2 Report

Comments and Suggestions for Authors

The wave model outlined in this manuscript suggests that during NREM sleep, energy is prepared in what the authors refer to as a "wave packet." As the system approaches the equilibrium zone between sleep and wakefulness, the interactions between NREM and REM waves result in a coherent superposition, ultimately leading to REM sleep. During REM sleep, a portion of the primed energy is released, followed by further wave packet evolution during subsequent NREMS episodes. Across successive sleep cycles, there is a gradual dissipation of energy associated with state instability, ultimately bringing the system closer to a more stable wakeful state. Consequently, longer and more intense REM sleep episodes increase the likelihood of spontaneous awakening, as they contribute to a more effective reduction of state instability. Understanding these dynamics enables researchers to make predictions about sleep architecture. The manuscript is exceptionally well-written; however, I have a few suggestions and questions as follows:

a) It would be helpful if the authors could provide a schematic representation illustrating the relationship between state instability and the transitions between different sleep states, which could enhance readers' comprehension.

b) Could the authors elaborate on the practical applications of this wave model? For instance, does it have the potential to predict drug-induced sleep disruptions or other sleep-related phenomena?

c) Is it possible that this model could predict the occurrence of hypersomnolence following substance abuse? An exploration of this topic would be insightful.

d) It would be valuable if the manuscript addressed any limitations associated with the study or the model itself. This could provide a more comprehensive understanding of the model's scope and applicability.
